# Deoxyribose and deoxysugar derivatives from photoprocessed astrophysical ice analogues and comparison to meteorites

Michel Nuevo [1,2], George Cooper[3] & Scott A. Sandford [1]

Sugars and their derivatives are essential to all terrestrial life. Their presence in meteorites, together with amino acids, nucleobases, amphiphiles, and other compounds of biological importance, may have contributed to the inventory of organics that played a role in the emergence of life on Earth. Sugars, including ribose (the sugar of RNA), and other sugar derivatives have been identified in laboratory experiments simulating photoprocessing of ices under astrophysical conditions. In this work, we report the detection of 2-deoxyribose (the sugar of DNA) and several deoxysugar derivatives in residues produced from the ultraviolet irradiation of ice mixtures consisting of $H_2O$ and $CH_3OH$. The detection of deoxysugar derivatives adds to the inventory of compounds of biological interest that can form under astrophysical conditions and puts constraints on their abiotic formation pathway. Finally, we report that some of the deoxysugar derivatives found in our residues are also newly identified in carbonaceous meteorites.

[1] NASA Ames Research Center, MS 245-6, Moffett Field, CA 94035, USA. [2] BAER Institute, NASA Research Park, MS 18-4, Moffett Field, CA 94035, USA. [3] NASA Ames Research Center, MS 239-4, Moffett Field, CA 94035, USA. Correspondence and requests for materials should be addressed to M.N. (email: michel.nuevo-1@nasa.gov)

Sugars (monosaccharides) and their derivatives are ubiquitous in nature and essential to biological processes in all terrestrial life. Sugars themselves play several roles in biology. For example, they are used as structural backbones in RNA (ribose) and DNA (deoxyribose), as well as cell walls in plants (cellulose). They also serve as energy sources (glucose) or energy storage (glycogen and starch)[1,2]. Structurally, sugars are carbohydrates of general formula $C_nH_{2n}O_n$, and the simplest sugars are defined as aldehydes (aldoses) or ketones (ketoses) containing at least two hydroxyl groups[3]. Sugar derivatives, collectively referred to as polyols, include sugar alcohols and sugar acids. In the case of aldoses, such sugar alcohols and acids are sugars in which the terminal aldehyde group is replaced by a primary alcohol or a carboxylic acid group, respectively. Deoxy variants of sugars, sugar alcohols, and sugar acids, are sugar derivatives that are lacking one or more hydroxyl groups with respect to their corresponding parent compounds (e.g., deoxyribose vs. ribose).

The smallest ketose sugar, dihydroxyacetone, as well as several sugar acids and sugar alcohols—with up to 6 carbon atoms—have been detected in several carbonaceous chondrites including Murchison and Murray, and shown to be extraterrestrial in origin[4,5]. The presence of sugar derivatives in primitive meteorites, together with other compounds of biological interest such as amino acids[6,7], nucleobases[8,9], and amphiphiles[10,11] is consistent with a scenario in which a significant fraction of the inventory of compounds from which biological processes started on the primitive Earth may have been delivered via comets, meteorites, and interplanetary dust particles (IDPs)[12,13].

Over the last 25 years, laboratory experiments simulating the photo-irradiation or particle bombardment of mixtures of astrophysical ice analogs at low temperatures (10–80 K) have shown that organic compounds can form under astrophysical, non-biological conditions. In particular, the ultraviolet (UV) irradiation of ice mixtures consisting of $H_2O$, $CH_3OH$, $CO$, $CO_2$, $CH_4$, and/or $NH_3$ leads to the formation of compounds of astrobiological interest such as amino acids[14–16], nucleobases[17,18], functionalized polycyclic aromatic hydrocarbons (PAHs)[19,20], amphiphiles[21], as well as urea, hydantoin, and small aldehydes[22–24]. A variety of organic compounds have also been shown to form at 5 K when ice mixtures containing $CH_3OH$ were UV irradiated or bombarded with energetic electrons[25,26]. Finally, a large variety of sugar alcohols, sugars (including ribose, the sugar of RNA), and sugar acids with up to 5 carbon atoms have been identified in residues produced from the UV irradiation of ice mixtures containing $H_2O$ and $CH_3OH$ at low temperatures[27,28]. In contrast, only 1,3-propanediol (3C deoxysugar alcohol) was unambiguously identified in a laboratory residue[28], while several deoxysugar acids have been found in meteorites[4,5]. The previous non-detection of deoxysugar derivatives in laboratory residues led to the conclusion that in such photo-irradiation experiments, sugar derivatives were formed via a formose-type reaction pathway[28].

The present study is based on the analysis of 5 independent residues produced from the UV irradiation of $H_2O$:$CH_3OH$ and $H_2O$:$^{13}CH_3OH$ ice mixtures in relative proportions of 2:1 at 12 K. The results obtained for these 5 residues are consistent with 12 other residues produced from $H_2O$:$CH_3OH$ ice mixtures in relative proportions 2:1 and 5:1 that were originally analyzed for the presence of other sugar derivatives. The 2:1 ratio between $H_2O$ and $CH_3OH$ ices corresponds to the upper limit of the [$H_2O$]/[$CH_3OH$] ratio observed in cold interstellar clouds[29,30]. In two of the 5 experiments prepared for the present study, methanol was isotopically labeled with $^{13}C$ in order to verify that the compounds detected in the residues were not due to

contamination. All residues were analyzed with gas chromatography coupled with mass spectrometry (GC-MS) using three distinct derivatization methods and temperature programs (see Methods). Most compounds were identified via their (+)-butyl/ trifluoroacetyl (but/TFA) derivatives by comparison with commercial standards prepared in the same manner as the samples, while some were identified via their tert-butyldimethylsilyl (t-BDMS) and/or trimethylsilyl (TMS) derivatives (see Methods). GC-MS analysis of residues with the (+)-butanol/TFAA method allowed for the enantiomeric separation of most of the sugar derivatives, thus allowing for an additional assessment of possible contamination in the samples.

## Results

**Identification of deoxyribose and deoxysugar derivatives.** All the residues produced in this work contained a wide variety of sugars, sugar alcohols, and sugar acids, with a similar distribution to previous laboratory ice UV irradiation studies[27,28]. In addition to these common sugar derivatives, we identified a number of deoxysugars, deoxysugar alcohols, and deoxysugar acids, i.e., compounds in which one or more carbon atoms is not bonded to a hydroxyl (OH) group relative to the corresponding canonical parent sugars, sugar alcohols, and sugar acids, respectively. These include the 5-carbon (5C) deoxysugar 2-deoxyribose ($C_5H_{10}O_4$) (the structural backbone of DNA), as well as the 3C and 4C deoxysugar alcohols 1,2-propanediol ($C_3H_8O_2$), 1,3-propanediol ($C_3H_8O_2$), 2-methyl-1,3-propanediol ($C_4H_{10}O_2$), 1,2,3-butanetriol ($C_4H_{10}O_3$), and 1,2,4-butanetriol ($C_4H_{10}O_3$). The molecular structures of all the compounds identified in our residues are shown in Supplementary Fig. 1. This is the first definitive identification of a deoxysugar in laboratory ice photolysis residues. Only one other deoxysugar derivative, namely, 1,3-propanediol (deoxysugar alcohol, also present in our residues) was unambiguously detected in one other laboratory residue in a previous study, together with tentative identifications of 2-methylglycerol (deoxysugar alcohol) and 2-methylglyceric acid (deoxysugar acid) (not detected in our residues)[28].

The GC-MS chromatograms and mass spectra supporting the identification of 2-deoxyribose and several deoxysugar alcohols in our residues, produced from the UV irradiation of $H_2O$:$CH_3OH$ (2:1) and $H_2O$:$^{13}CH_3OH$ (2:1) ice mixtures, are shown in Fig. 1 (deoxyribose), Fig. 2 (4C deoxysugar alcohols), and Supplementary Fig. 2 (3C deoxysugar alcohols). The peaks eluting at retention times around 57.0 and 57.3 min in Fig. 1a are tentatively assigned to the D and L enantiomers of 2-deoxyxylose, respectively. This tentative identification is based on the mass spectra of these peaks (Supplementary Fig. 3), which are very similar to that of 2-deoxyribose (Fig. 1b), and because 2-deoxyxylose is the only possible diastereomer of 2-deoxyribose. Although its identification is not formally confirmed yet for lack of a standard, the mass spectra of diastereomers of polyols are usually identical with the derivatization methods used in the present work.

Typical residues were produced from $H_2O$:$CH_3OH$ (2:1) and $H_2O$:$^{13}CH_3OH$ (2:1) ice mixtures in which totals of 1.23 mmol of $H_2O$ and 0.61 mmol of $CH_3OH$/$^{13}CH_3OH$ were co-deposited at rates of about 1.7 μmol min$^{-1}$, while being simultaneously irradiated for 17–19 h with photon doses of 0.35–0.39 photons molecule$^{-1}$. The deoxysugar derivatives, as well as ribose for comparison, identified in all regular (i.e., mostly $^{12}C$) and $^{13}C$-labeled residues are summarized in Table 1. Note that the abundances given for 1,2-propanediol and 1,3-propanediol are lower limits, because these compounds are very volatile and a

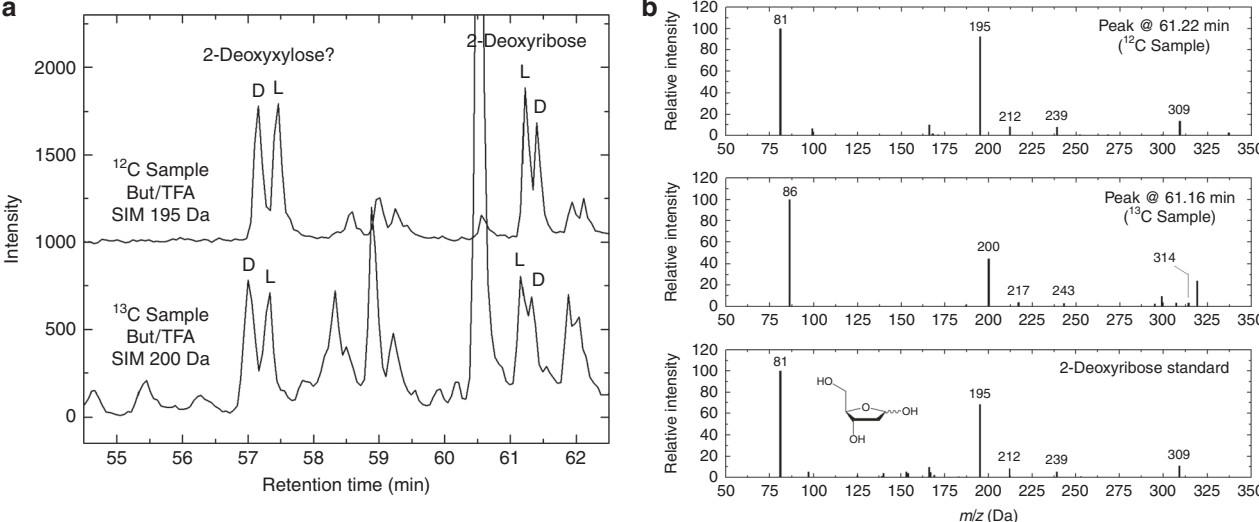

**Fig. 1** Identification of 2-deoxyribose in ice photolysis residues. **a** Single-ion monitoring (SIM) chromatograms of residues produced from the UV irradiation of $H_2O$:$CH_3OH$ (2:1) ($^{12}C$ sample, $m/z = 195$ Da) and $H_2O$:$^{13}CH_3OH$ (2:1) ($^{13}C$ sample, 200 Da) ice mixtures after derivatization with (+)-2-butanol/TFAA. The peaks around 57.0 and 57.3 min in both chromatograms are tentatively assigned to 2-deoxyxylose (see Supplementary Fig. 2). Intensities are offset for clarity. **b** From top to bottom, mass spectra of the peaks assigned to 2-deoxy-L-ribose in the regular residue, the $^{13}C$-labeled residue, and a standard of 2-deoxyribose. The molecular structure of 2-deoxyribose is shown without derivatization. Assignments of the fragments in the mass spectra can be found in Supplementary Table 1

non-negligible fraction of them may have sublimed away during the ice warm-up phase and/or the preparation of the samples for GC-MS analysis (see Methods). In the residue whose chromatogram is shown as the bottom trace of Fig. 1a, the abundances of 2-deoxyribose ($^{13}C$-labeled D + L enantiomers) ranged from 334 to 3855 pmol, which corresponds to production yields of $3 \times 10^{-6}$ to $3.2 \times 10^{-5}$ from the starting $^{13}C$-methanol.

Residues produced in these experiments probably contain several other deoxysugar derivatives, as supported by the presence of several peaks in the GC-MS chromatograms that display fragments with masses consistent with such compounds. These include an unidentified 4C deoxysugar acid that was tentatively assigned to 3,4-dihydroxybutyric acid by comparison of the mass spectrum of its *t*-BDMS derivative in the samples with the NIST database (Supplementary Fig. 4). However, their formal identification was difficult to assure because the corresponding standards were unavailable (only some of the isomers of the 4C deoxysugar acids were available as standards; see Methods), and because their high retention times may indicate that they are present as dimers or higher oligomers rather than monomers. The assignment to a deoxysugar acid is based on the presence in its mass spectrum of the fragment ion at 267 Da (270 Da for $^{13}C$-labeled compounds; Supplementary Fig. 4), which is due to the loss of a carboxyl/butyl group. In addition, the 6C deoxysugars 2-deoxygalactose and 2-deoxyglucose were searched for in the same residues, but the low signal-to-noise for the corresponding peaks could neither confirm nor disprove their presence in the residues.

Sugar derivatives are also known to form in detectable amounts in residues produced from the UV irradiation of simple ice mixtures containing $H_2O$ and $CH_3OH$ at low temperature[27,28]. In the five residues that were analyzed in the present study, we also detected several sugar alcohols, sugars, and sugar acids (in decreasing order of abundances), including ribose (racemic) (Fig. 3) with abundances of 237–2467 pmol (920 pmol in the $^{13}C$-labeled residue shown in Fig. 1) (Table 1).

**Production of indigenous compounds vs. contamination.** The 2-deoxyribose and most of the other deoxysugar derivatives identified in the residues produced from ices containing $^{13}C$-methanol were found to be isotopically labeled with $^{13}C$, as shown in their mass spectra (Figs. 1 and 2). The only exception comes from 1,2-propanediol and 1,3-propanediol, which could not be found in $^{13}C$-labeled residues (Supplementary Fig. 2), probably because of their high volatility and the fact that abundances in these $^{13}C$-labeled residues were lower than in regular ($^{12}C$) residues.

The peaks assigned to the enantiomers of 2-deoxyribose (Fig. 1) and those tentatively assigned to those of 2-deoxyxylose (Supplementary Fig. 3) show that these compounds were formed in racemic mixtures in both regular and $^{13}C$-labeled residues. This indicates that the compounds identified in our residues result from the UV photoprocessing of the ices, and rules out the possibility of exogenous contamination. Indeed, not only is 2-deoxy-L-ribose a very rare compound on Earth as biological DNA exclusively uses the D enantiomer, but biological processes also tend to enrich biomolecules in $^{12}C$.

The GC-MS chromatograms of three independent control samples, one in which no ice was deposited and the substrate irradiated, one in which only $H_2O$ was deposited and irradiated, and one in which an $H_2O$:$^{13}CH_3OH$ (2:1) ice mixture was deposited but not irradiated (Supplementary Fig. 5 and Methods), show small amounts of contaminant $^{12}C$-2-deoxy-D-ribose (Supplementary Fig. 5a), but no contaminant $^{12}C$-ribose. D-Arabinose, which elutes right after D-ribose (Supplementary Fig. 5b), is produced in our experiments and found to be present in all residues. The level of contamination from $^{12}C$-2-deoxy-D-ribose varies slightly from one control sample to another, with an estimated contribution smaller than 9% of the amounts of 2-deoxyribose measured in the residues. This suggests that such contamination may come from the solvents, derivatization agents, and/or glassware. The contribution of contaminant 2-deoxyribose

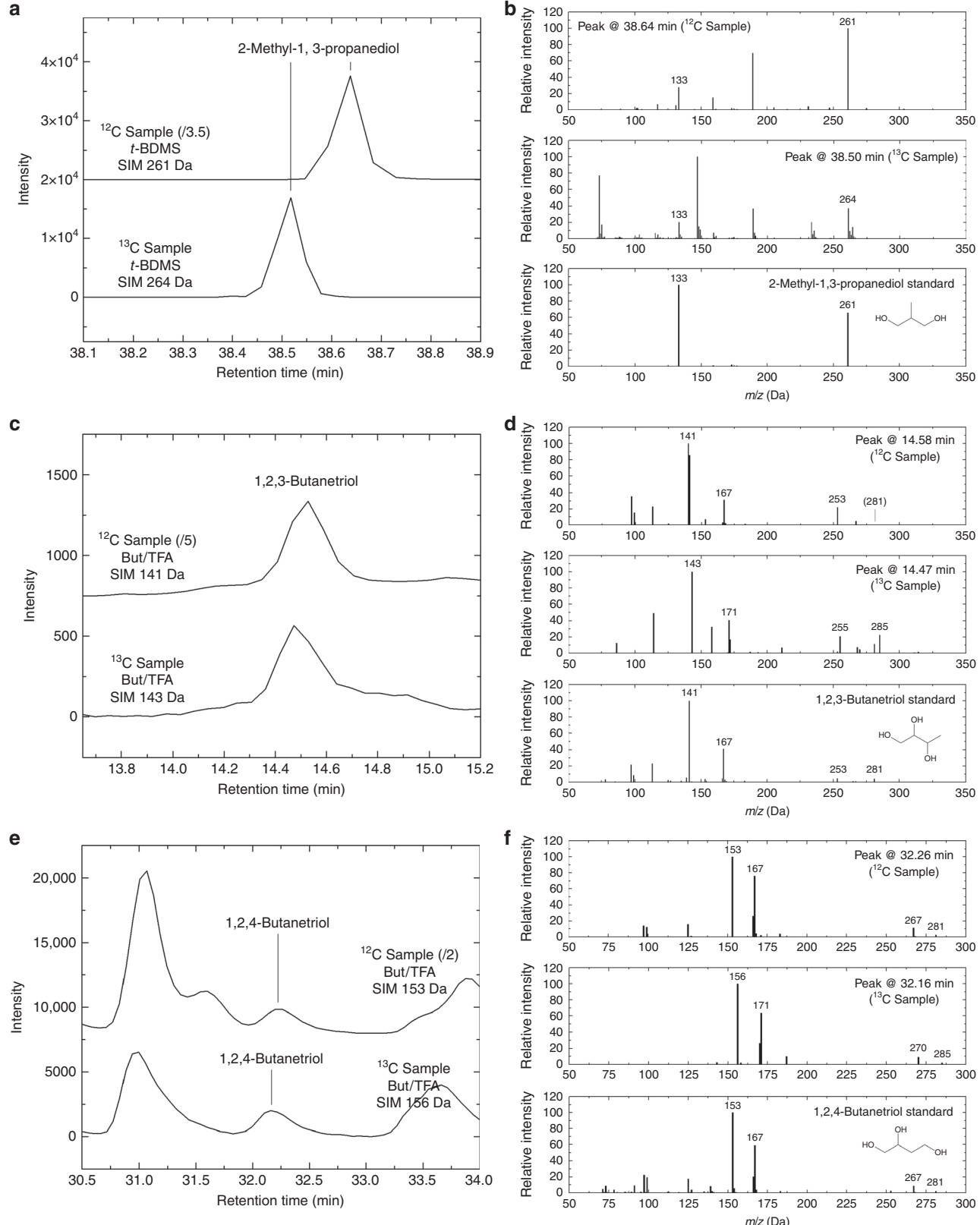

was taken into account for the produced abundances in the residues (Tables 1 and 2).

## Discussion

The formation mechanisms for deoxysugar derivatives under our experimental conditions simulating ice photochemistry in cold astrophysical environments are difficult to determine, due to the stochastic nature of the chemical processes taking place in the ice matrix. Indeed, the energy of the incident UV photons is sufficient to break chemical bonds and ionize species, resulting in the release of H atoms together with OH, $CH_3$, $CH_2OH$, and $CH_3O$ radicals and ions from the photolysis of $H_2O$ and $CH_3OH$. At 12

**Fig. 2** Identification of the 4C deoxysugar alcohols 2-methyl-1,3-propanediol, 1,2,3-butanetriol, and 1,2,4-butanetriol in ice photolysis residues. **a** SIM chromatograms of residues produced from the UV irradiation of $H_2O$:$CH_3OH$ (2:1) ([12]C sample, $m/z = 261$ Da) and $H_2O$:[13]$CH_3OH$ (2:1) ([13]C sample, 264 Da) ice mixtures after derivatization with MTBSTFA. Intensities are offset for clarity. **b** From top to bottom, mass spectra of the peaks assigned to 2-methyl-1,3-propanediol in the regular residue, the [13]C-labeled residue, and a standard of 2-methyl-1,3-propanediol. The difference in relative intensities for the peaks at 133 and 261 Da suggests a coelution with an unidentified compound with similar mass fragments. **c** SIM chromatograms of the same regular (141 Da) and [13]C-labeled (143 Da) residues after derivatization with (+)-2-butanol/TFAA. Intensities are offset for clarity. **d** From top to bottom, mass spectra of the peaks assigned to 1,2,3-butanetriol in the regular residue, the [13]C-labeled residue, and a standard of 1,2,3-butanetriol. **e** SIM chromatograms of the same regular (153 Da) and [13]C-labeled (156 Da) residues after derivatization with (+)-2-butanol/TFAA. Intensities are offset for clarity. **f** From top to bottom, mass spectra of the peaks assigned to 1,2,4-butanetriol in the regular residue, the [13]C-labeled residue, and a standard of 1,2,4-butanetriol. Molecular structures are shown without derivatization. Assignments of the fragments in the mass spectra can be found in Supplementary Table 1

**Table 1 Deoxysugar derivatives identified in the ice photolysis residues (regular and [13]C-labeled)**

| Compounds[a] | Formulas | $R_t$ (min)[b] | Abundances in residues[c] (pmol) | Detected in meteorites? |
|---|---|---|---|---|
| *Deoxysugars* | | | | |
| 2-Deoxyribose | $C_5H_{10}O_4$ | 61.2, 61.4 | 217–3855 | Undetermined[k] |
| 2-Deoxyxylose[d] | $C_5H_{10}O_4$ | 57.0, 57.3 | 373–3636[e] | Undetermined[k] |
| *Deoxysugar alcohols* | | | | |
| 1,2-Propanediol[f] | $C_3H_8O_2$ | 9.9 | ≥8–375 | Yes[l,m] |
| 1,3-Propanediol[f,g] | $C_3H_8O_2$ | 36.9 | ≥19–27 | No |
| 2-Methyl-1,3-propanediol[g,h] | $C_4H_{10}O_2$ | 38.7 | ≤1038–3354[h] | No |
| 2-(Hydroxymethyl)-1,3-propanediol | $C_4H_{10}O_3$ | 30.9 | n.d. | Yes[l] |
| 1,2,3-Butanetriol | $C_4H_{10}O_3$ | 14.5 | 6–39 | No |
| 1,2,4-Butanetriol | $C_4H_{10}O_3$ | 32.2 | 35–50 | Yes[l] |
| *Deoxysugar acids* | | | | |
| 3,4-Dihydroxybutyric acid[i,j] | $C_4H_8O_4$ | 16.5 | — | Yes[n] |
| *Sugars* | | | | |
| Ribose | $C_5H_{10}O_5$ | 64.7, 65.0 | 237–2467 | No |

n.d., Not detected
[a]Compounds were detected using the (+)-2-butanol/TFAA derivatization method, unless otherwise stated
[b]Retention times ($R_t$) correspond to average values in the GC-MS chromatograms of the residues, or to standard chromatograms if compounds were not detected in the residues. Chiral compounds whose enantiomers are separated have two retention times
[c]Abundances for chiral compounds correspond to the sum for both enantiomers
[d]Tentatively identified by comparison of its mass spectrum with that of the 2-deoxyribose standard (see Fig. 1 and Supplementary Fig. 3)
[e]Abundances estimated based on the GC-MS detector response for the 2-deoxyribose standard
[f]Volatile compounds that may have been partially lost during the warm-up phase and/or the sample preparation. Abundances given thus correspond to lower limits
[g]Detected in samples derivatized with the MTBSTFA method
[h]Elutes with another unidentified compound with similar mass fragments, so abundances given are upper limits
[i]Tentatively identified by comparison with the NIST mass spectrometry library in samples derivatized with the BSTFA method
[j]May be present in its dimer form
[k]The presence of these compounds in meteorites is uncertain, and further analyses are required
[l]Detected in Murchison (Fig. 4)
[m]Detected in GRA 06100 (Fig. 4)
[n]Detected in Murchison and Murray[4]

K, such reactive species have very limited mobility, and radical recombination takes place between neighboring species, both at low temperature and during warm-up, resulting in the formation of a wide variety of new species which are not necessarily the most thermodynamically stable products. However, UV photons can also break newly formed species into smaller fragments, so that the pathway towards the formation of a given product is the result of multiple recombination and destruction reactions. Consequently, the number of photons required to form larger molecules is typically larger than the number of chemical steps required to go from reactants to products.

Under these conditions, all products identified in the final residues likely form via several distinct pathways rather than only one. Such processes have been studied in more detail for the formation of amino acids from the UV irradiation of ices under similar experimental conditions[31]. In addition, we would note that some photoproducts may be formed during the warm-up stage, after UV photolysis, from species formed at low temperature and trapped in the ices until the temperature is high enough

for them to be mobile and react. During the warm-up process, smaller, higher-volatility compounds may either sublime away or stay trapped in the residues, resulting in a loss of some of these photoproducts. Therefore, due to the stochastic nature of the processes taking place in UV irradiation experiments at low temperature and the subsequent warm-up to room temperature, the final abundances of the photoproducts recovered in the residues (at room temperature) can vary significantly from one sample to another (Table 1), even when experiments are performed under very similar experimental conditions.

Sugar derivatives have been shown to form abiotically via formose-type reactions from the oligomerization of formaldehyde in an aqueous solution in the presence of a basic catalyst and small amounts of one or more initiators such as glycolaldehyde, glyceraldehyde, dihydroxyacetone, or any larger sugar[32,33]. Formose-type reactions may also, in some cases, be initiated by UV light without any catalyst or initiator[34]. This mechanism was proposed to form the sugars (including ribose) and other sugar derivatives produced from the UV irradiation of one

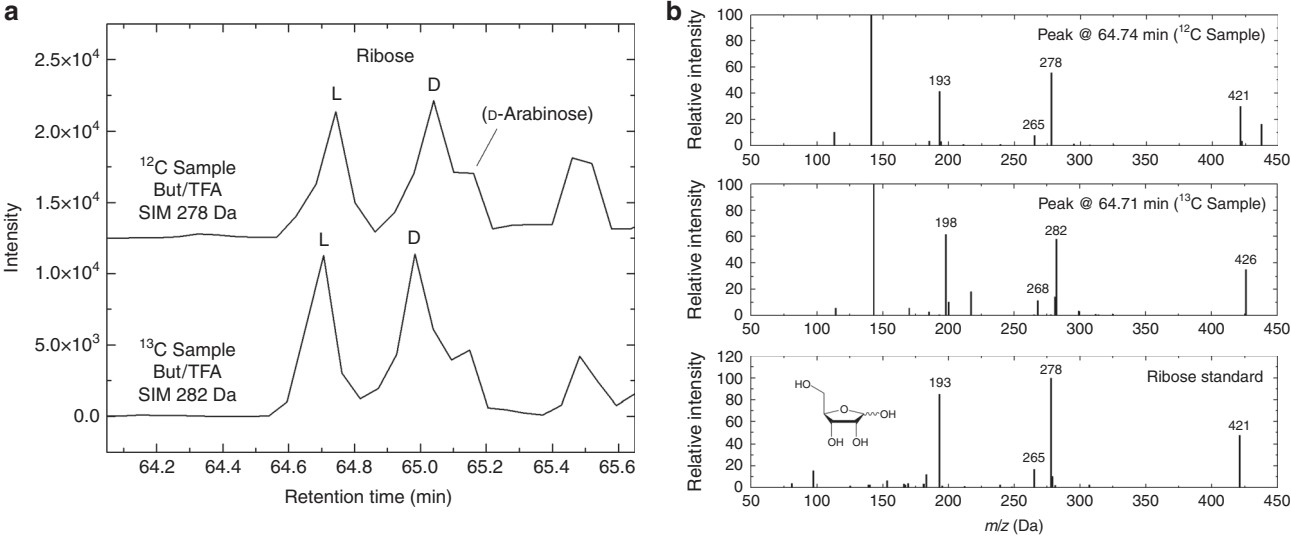

**Fig. 3** Identification of ribose in ice photolysis residues. **a** SIM chromatograms of residues produced from the UV irradiation of $H_2O:CH_3OH$ (2:1) ($^{12}C$ sample, $m/z = 278$ Da) and $H_2O:^{13}CH_3OH$ (2:1) ($^{13}C$ sample, 282 Da) ice mixtures after derivatization with (+)-2-butanol/TFAA. Intensities are offset for clarity. **b** From top to bottom, mass spectra of the peaks assigned to L-ribose in the regular residue, the $^{13}C$-labeled residue, and a standard of ribose. The molecular structure of ribose is shown without derivatization. Assignments of the fragments in the mass spectra can be found in Supplementary Table 1

**Table 2 Ratios between the abundances of 2-deoxyribose and ribose in the residues**

| Samples | Deoxyribose/ribose ratio (D + L)[a] | Deoxyribose/ribose ratio (L only)[a] |
|---|---|---|
| Regular residues | 0.15–3.33 | 0.23–4.45 |
| $^{13}C$-residues | 0.23–0.24 | 0.31–0.33 |

[a]Ratios were calculated from the total abundances of either both D + L enantiomers (after subtracting the contribution from contaminant $^{12}C$-2-deoxy-D-ribose, see Supplementary Fig. 5) or L enantiomers only.

$H_2O:^{13}CH_3OH:NH_3$ ice mixture in an independent study[28]. However, no experimental proof of such a mechanism was provided in that study, and the experimental conditions of formose-type reactions are quite different from those in ice irradiations. Moreover, although formaldehyde is one of the major photoproducts of methanol when subjected to UV irradiation[35], its formation requires several reaction steps from the starting methanol, so it forms together with a number of other, competing methanol photoproducts. However, once formed, formaldehyde may undergo a photo-induced oligomerization similar to a formose-type reaction process, leading to the formation of sugars and sugar derivatives. In addition, a formose-type mechanism is expected to yield sugars as the majority products, including branched compounds, but no deoxysugar derivatives[32,33]. The presence of several deoxysugar derivatives in our residues, including 2-deoxyribose with abundances sometimes higher than ribose, therefore indicates that other competing mechanisms are also involved in the formation of the identified photoproducts.

To the best of our knowledge, only two other abiotic synthesis of 2-deoxyribose have previously been reported in the literature, although neither describes a detailed formation mechanism. The first one involves the reaction between acetaldehyde ($C_2H_4O$), glyceraldehyde ($C_3H_6O_3$), and calcium oxide (CaO) at 50 °C in an aqueous solution, which leads to a production yield of 3% for 2-deoxyribose[36]. The use of formaldehyde ($H_2CO$) instead of glyceraldehyde led to a smaller yield. The second reported abiotic synthesis of 2-deoxyribose describes the photo-induced deoxygenation of the 5C sugars ribose and/or arabinose in solution, when mixed with a solution of $H_2O/D_2O$, $NaH_2PO_4 \cdot 2 \ H_2O$, KSCN, and NaSH $\cdot \ xH_2O$ at 37 °C, pH 7, and when subjected to the UV light emitted by Hg bulbs ($\lambda = 254$ nm)[37]. After 6 h of irradiation, it was found that more than half of the starting ribose/arabinose had been converted into photoproducts, among which 2-deoxyribose was identified using $^1H$-NMR analysis[37].

The ratios of the abundance of 2-deoxyribose to that of ribose in our residues span a wide range, i.e., 0.15–3.33 for D + L enantiomers and 0.23–4.45 for L enantiomers only (Table 2). While a more detailed study is necessary to formally determine the mechanism(s) involved in the formation of deoxysugar derivatives under our experimental conditions, these ratios do not point to any particular mechanism for the formation of 2-deoxyribose. Possibilities for 2-deoxyribose formation therefore include a bottom-to-top mechanism involving smaller intermediates than 5C sugar derivatives (such as described in ref. [36]), a photo-induced deoxygenation of ribose and/or arabinose previously made from the ice irradiation (such as described in ref. [37]), or a combination of both pathways, as would be expected from the stochastic nature of the processes involved in the formation of these compounds. In any case, our results do not support formation mechanisms involving only formose-type reactions[32,33].

Deoxysugar derivatives were also found to be present in carbonaceous chondrites such as Murchison in the form of 4C–6C deoxysugar acids[4,5]. The wide variety of deoxysugar acids present in detectable quantities in the Murchison and Murray meteorites mirrors the variety of canonical sugar acids found in the same meteorites. Indeed, sugar acids were found to be the second most abundant family of polyols present in meteorites after sugar alcohols, while only one sugar, dihydroxyacetone, a ketose sugar, was detected[4]. An analysis of Murchison and Graves Nunataks (GRA) 06100 meteorite samples for this study shows that small

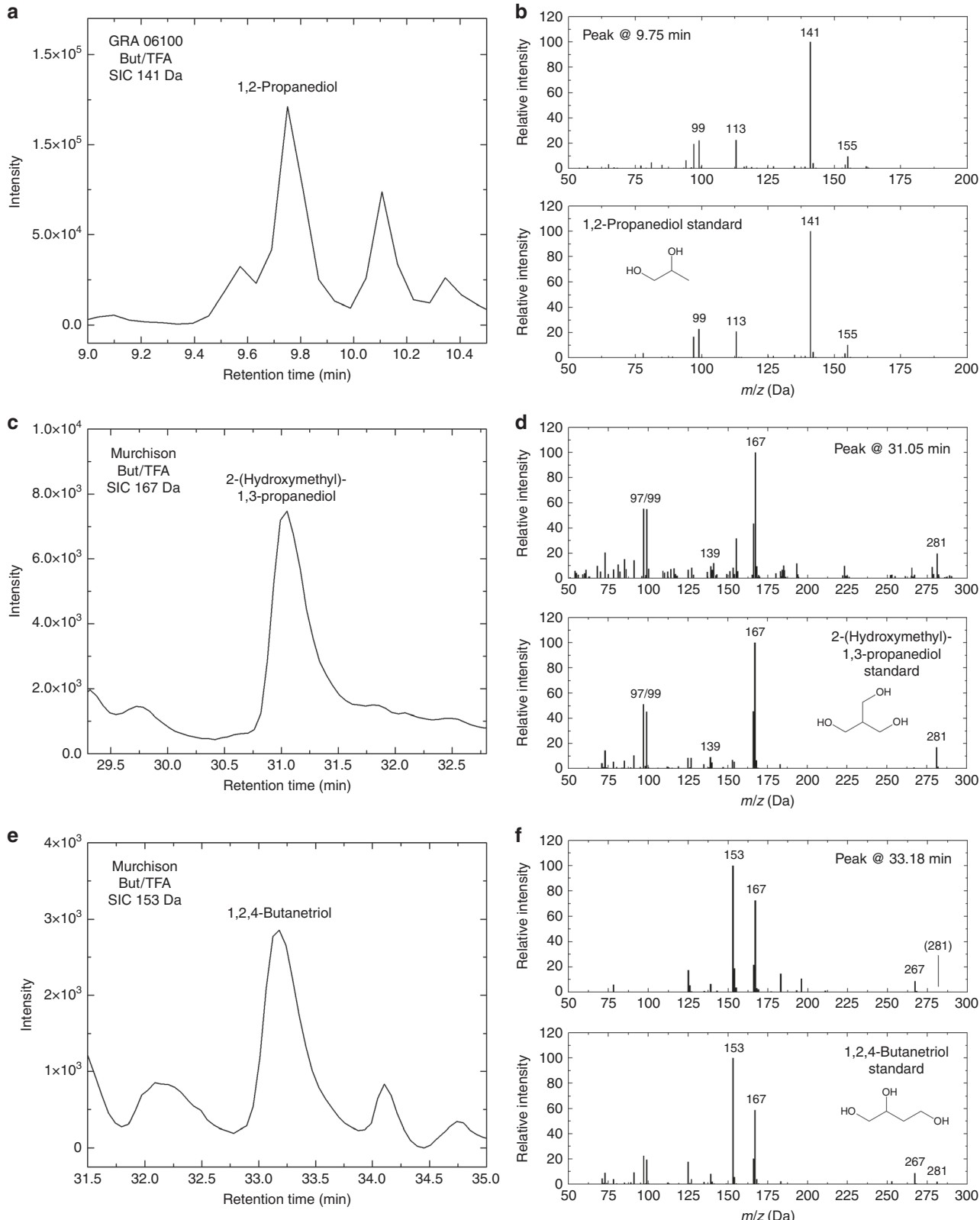

**Fig. 4** Identification of three deoxysugar alcohols in meteorites. **a** Single-ion chromatogram (SIC) of a sample from GRA 06100 ($m/z = 141$ Da) derivatized with (+)-2-butanol/TFAA. **b** Mass spectrum of the peak assigned to 1,2-propanediol, compared with the mass spectrum of a standard of 1,2-propanediol. **c** SIC of a sample from Murchison (167 Da) derivatized with (+)-2-butanol/TFAA. **d** Mass spectrum of the peak assigned to 2-(hydroxymethyl)-1,3-propanediol, compared with the mass spectrum of a standard of 2-(hydroxymethyl)-1,3-propanediol. **e** SIC of the same Murchison sample (153 Da). **f** Mass spectrum of the peak assigned to 1,2,4-butanetriol, compared with the mass spectrum of a standard of 1,2,4-butanetriol. Deoxysugar alcohols only accept TFAA derivatization under the present conditions, i.e., only O–TFA bonds are formed. Molecular structures are shown without derivatization. Assignments of the fragments in the mass spectra can be found in Supplementary Table 1

deoxysugar alcohols are also present in meteorites (Fig. 4): 1,2-propanediol (3C) was found in Murchison and GRA 06100, but not its isomer 1,3-propanediol, while 2-(hydroxymethyl)-1,3-propanediol (4C, branched) and 1,2,4-butanetriol (4C) were both found in Murchison. The presence of a small compound such as 1,2-propanediol in GRA 06100 is surprising, as the parent body of this meteorite is believed to have experienced temperatures as high as 600 °C[38]. However, the pyrolysis (>500 °C) of 1,2-propanediol showed that this compound is stable at very high temperatures[39], which supports its presence in GRA 06100. Larger deoxysugars such as 2-deoxyribose and 2-deoxyglucose were also searched for in these meteorites, but their presence could not be unambiguously confirmed (larger sample sizes might be needed). Among all these compounds, only 1,2-propanediol and 1,2,4-butanetriol were also found in our residues (Table 1).

This observation does not rule out a formation of sugars and deoxysugars via ice photochemistry in cold astrophysical environments along with their alcohol and acid derivatives, although it seems to indicate that sugars and deoxysugars may not be as stable as their alcohol and acid counterparts in the physical and chemical environments they experience between their formation and their incorporation into meteorite parent bodies, i.e., asteroids and comets. The absence of sugars—except one—and lack of definitive detection of deoxysugars in the meteorites analyzed so far may also be due to their very low abundances and/or the result of their reduction or oxidation into the corresponding alcohol and acid derivatives, respectively, in the astrophysical environments where they formed or after aqueous alteration in asteroids and comets. Future analyses of larger meteoritic samples might be more conclusive in the hunt for compounds such as deoxyribose, in particular because deoxyribose is more stable than ribose. In any case, it indicates that the sugars and deoxysugars essential to terrestrial life may have been delivered to the primitive Earth predominantly in the form of their alcohol and acid derivatives rather than in their aldose/ketose form.

The efficient production of sugar derivatives[27,28] and deoxysugar derivatives (this work) from the UV irradiation of astrophysical ice analogs supports scenarios in which ice photochemistry plays an important role in the formation of the organics that are detected in carbonaceous meteorites. In particular, the presence of both sugar and deoxysugar derivatives in laboratory residues and meteorites compounds favors ice photochemistry over a formose-type reaction mechanism for their formation, as deoxysugar derivatives have not been reported as products of the formose reaction[32,33]. However, the formation mechanism and meteoritic distribution of these compounds need to be studied in more detail.

Organic compounds of important astrobiological interest that have been found in primitive meteorites, and which include amino acids[6,7], nucleobases[8,9], amphiphilic compounds[10,11], as well as sugar and deoxysugar derivatives[4,5], were delivered to the primitive Earth via asteroids and comets[12,13]. Though terrestrial processes must also have contributed to the emergence of life on our planet over 3.8 billion years ago[40], those meteoritic organics were available and may have played a role in the first biological processes. In the case of deoxysugar derivatives, larger (and different) meteorite samples may be more definitive as to their presence in extraterrestrial environments. The formation of complex organics in astrophysical environments and the delivery of compounds of biological importance to telluric planets are believed to be universal events that may have occurred elsewhere in the Universe.

## Methods

**Irradiation of ices at low temperature.** Gas mixtures were prepared in a glass line (background pressure: ~$10^{-6}$ mbar) by mixing the vapors of $H_2O$ (liquid; purified to 18.2 MΩ cm by a Millipore Direct-Q UV 3 device) and either $CH_3OH$ (liquid; Aldrich, HPLC grade, ≥99.9% purity) or $^{13}CH_3OH$ (liquid; Cambridge Isotope Laboratories, Inc., 99.9% $^{13}C$) in a glass bulb (volume: 2.09 L). All liquids were freeze-pump-thawed at least three times prior to mixing of their sublimation gases, in order to remove excess dissolved gases. Ratios between mixture components were determined by their partial pressure (±0.05 mbar). $H_2O:CH_3OH$ and $H_2O:^{13}CH_3OH$ gas mixtures in relative proportion 2:1 were prepared for this work, for total pressures of ~27 mbar in each bulb.

These gas mixtures were injected into a vacuum cryogenic chamber evacuated to a few $10^{-8}$ mbar, and deposited onto a piece of aluminum (Al) foil (prebaked to 500 °C), used as a substrate and attached to a cold finger cooled to 12 K by a closed-cycle helium cryocooler. Deposited mixtures formed amorphous ice films which were simultaneously exposed to UV irradiation for 17–19 h. The UV source used was a microwave-powered $H_2$-discharge UV lamp (Opthos; $H_2$ pressure: 0.1 mbar) which emits UV photons at 121.6 nm (Lyman α) and a continuum centered around 160 nm[41], with a flux of ~$10^{15}$ photons cm$^{-2}$ s$^{-1}$ (estimate)[42]. Such a light source simulates the UV radiation field in the dense ISM and around protostars[43–45]. In this work, the UV doses received by each sample ranged from 0.35 to 0.39 photon molecule$^{-1}$. Assuming a photon flux of $8 \times 10^7$ photons cm$^{-2}$ s$^{-1}$ for photons with energies higher than 6 eV in the diffuse ISM[46], and fluxes at least 3 orders of magnitude lower in dense interstellar clouds[47,48], our experiments correspond to a UV irradiation of ices of about $5 \times 10^4$ yr in diffuse media, and >$5 \times 10^7$ yr in denser media. Note that ices are expected to be present mostly in cold, dense molecular clouds, so that the UV dose received by ices in one of our experiments is relevant to the dose ice-coated grains receive during the whole lifetime of a molecular cloud. In addition, such ice-coated grains are expected to experience UV doses up to 100 times higher in the solar nebula[49], so that one experiment would correspond to a full cycle during which a grain travels out of and back to the protosolar disc midplane. After simultaneous deposition and irradiation, samples were warmed under static vacuum to room temperature at about 0.75 K min$^{-1}$, at which time each residue covering the substrate was taken with 200 μL of $H_2O$ and transferred into a clean, prebaked (500 °C) vial for further analysis.

**Analysis of laboratory residues at room temperature.** Each $H_2O$-dissolved residue was divided into smaller aliquots to be analyzed with three different GC-MS methods. Before any derivatization, each aliquot was dried in a desiccator under vacuum for 1–2 h. The three derivatizations methods used in this work are: (+)-butanol/TFAA (identification and separation of enantiomers), BSTFA (identification of sugar derivatives containing 3 or more derivatizable groups), and MTBSTFA (identification of smaller deoxysugar derivatives that contain only two OH groups) derivatizations. Each derivatization method is described in detail in the Supplementary Methods section.

Sugar derivatives in residues were identified by comparison of both their retention times and mass spectra with commercial standards. These standards were purchased from Sigma-Aldrich, unless otherwise stated, and include: (1) deoxysugars: 2-deoxy-D-ribose ($C_5H_{10}O_4$; 97%), 2-deoxy-L-ribose ($C_5H_{10}O_4$; 97%), 2-deoxy-D-galactose ($C_6H_{12}O_5$; 98%), and 2-deoxy-D-glucose ($C_6H_{12}O_5$; 99%); (2) deoxysugar alcohols: 1,2-propanediol ($C_3H_8O_2$; Alfa Aesar, >98%), 1,3-propanediol ($C_3H_8O_2$; Tokyo Chemical Industry Co., >98.0%), 2-methyl-1,3-propanediol ($C_4H_{10}O_2$; Tokyo Chemical Industry Co., >98.0%), 1,2,3-butanetriol ($C_4H_{10}O_3$; Aldrich$^{CPR}$), (±)-1,2,4-butanetriol ($C_4H_{10}O_3$; 95%), and 2-(hydroxymethyl)-1,3-propanediol ($C_4H_{10}O_3$; 97%); and (3) deoxysugar acids: the four 2,3-dihydroxybutyric acid isomers (2R,3R; 2S,3S; 2R,3S; and 2S,3R; $C_4H_8O_4$; sodium salts, 97%).

**Control experiments.** In addition to the regular experiments of UV irradiation of ices, three control samples were prepared: (1) a piece of Al foil with no deposited ices that was exposed to UV irradiation, (2) a pure $H_2O$ ice (no methanol) deposited and exposed to UV irradiation, and (3) an $H_2O:^{13}CH_3OH$ (2:1) ice mixture deposited but not subjected to UV irradiation. These control experiments were carried out and analyzed with the (+)-2-butanol/TFAA method following the same procedure as for all other experiments.

**Analysis of meteorite samples.** The Murchison meteorite (CM2 chondrite) sample was obtained from the Center for Meteorite Studies, Arizona State University, Tempe, AZ. The GRA 06100 meteorite (CR2) sample was obtained from the Antarctic Meteorite Collection at NASA Johnson Space Center, Houston, TX. The samples of both meteorites were extracted in water, and the organic compounds were fractionated following procedures identical to those described previously[50]. Fractions containing neutral compounds, sugar alcohols, etc., and weak

acids were derivatized for GC-MS analysis as described above. The amounts of Murchison and GRA 06100 meteorites analyzed correspond to 66 and 50 mg, respectively.

## Data availability

Data from this manuscript will be made available on the NASA PubMed Central repository (https://www.nasa.gov/open/researchaccess/pubspace). This includes raw data as well as figures in the form of image files for both the main manuscript and the Supplementary Information. Alternatively, raw data and figure image files are available on request from the corresponding author M.N.

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

## Acknowledgements

We thank R.L. Walker (NASA Ames, retired) for technical support and A.C. Rios (NASA Ames) for useful comments on the manuscript. This work was supported by the National Aeronautics and Space Administration through the NASA Exobiology Program and the NASA Astrobiology Institute under Cooperative Agreement Notice NNH13ZDA017C issued through the Science Mission Directorate.

## Author contributions

M.N. prepared the laboratory samples (residues) and performed GC-MS analyses with BSTFA derivatization. G.C. performed GC-MS analysis of the residues with (+)-butanol/TFAA and MTBSTFA derivatizations. M.N. and G.C. analyzed the data to identify the compounds in the residues. S.A.S. helped for the interpretation of the results and their astrobiological implications. M.N. wrote the paper with inputs from G.C. and S.A.S.

## Additional information

**Competing interests:** The authors declare no competing interests.

