## [Peer Review File · Nature Communications]

Reviewer #1 (Remarks to the Author):

The manuscript entitled 'Deoxyribose and deoxysugar derivatives in residues produced from photoprocessed astrophysical ice analogues and comparison to meteorites' deals with the formation of 2-deoxyribose and other deoxysugar derivatives in residues by ultraviolet irradiation of ice mixtures consisting of H₂O and CH₃OH.

Overall this is a very interesting study of broad interest.

Here are my comments:

I recommend to remove the statement, that the 'organics that triggered life are of extraterrestrial origin' in the second sentence of the abstract paragraph. The question would be, why only on extraterrestrial objects and not on earth itself. There has been a recent publication about the formation of sugars triggered by Schreibersite, which is commonly found in meteorites Pallmann, S. et al. Schreibersite: an effective catalyst in the formose reaction network. *New J. Phys.* 20, 055003 (2018). This is a plausible mechanism, that sugars are formed on earth.

It would be very good to provide a high-resolution MS/MS spectrum of the potentially formed deoxyribose for absolute confirmation.

Is there a chance to give more insights in the formation mechanism? Methanol has to be oxidized in the reaction. How does this work. Is there residual oxygen or is it a dehydration reaction that is triggered by a surface reaction under UV irradiation. It is interesting, that apparently the selectivity to form sugars starting from methanol is better than starting with formaldehyde. The question is also, if formaldehyde polymerizes under these conditions or if it is still molecular formaldehyde?

All experiments performed are well described.

Reviewer #2 (Remarks to the Author):

This manuscript reports the detection of 2-deoxyribose and several deoxysugar derivatives in room-temperature residues produced from the VUV-irradiation of ice mixtures containing water and methanol (or ¹³C-labeled methanol). This manuscript is well written, the data is clearly presented, and the identification of deoxysugar compounds appear sound (mass spectra comparison to reference standards is convincing and the use of isotopic labels helps rule out the possibility that products are the result of contamination). The detection of 2-deoxyribose (along with the previous detection of ribose by Meinert et al.) in these reactions is highly significant since these organic

compounds are critical to the structure of DNA (and RNA) and would have important implications to the fields of cosmochemistry and origin of life/astrobiology. This is an important paper and should be published in Nature Communications.

I do have a few minor points that I hope the authors could address:

1. Deoxysugar acids appear to be diverse in carbonaceous chondrites in Cooper et al. (Nature 2001). Yet only one deoxysugar acid (3,4-dihydroxybutyric acid; Table 1) was identified in residues of VUV-irradiated H₂O:12 or 13CH₃OH ice mixtures in this report. Were other deoxysugar acids (4C, 5C, and 6C) searched for in these residues? 2-Methylglyceric acid, 2,4-dihydroxybutyric acid, 2,3-dihydroxybutyric acid, 2-deoxypentonic acids, 2-deoxyhexonic acids, and 3-deoxyhexonic acids were previously identified in Murchison and Murray meteorites by Cooper et al. If these compounds were searched for and not found in astrophysical ice analogs, what could be some possible explanations for the absence of these deoxysugar acids?

It is interesting to note that many of the sugar alcohols and sugar acids (3C, 4C, and 5C) in Murchison and Murray meteorites identified by Cooper et al. were also identified in the organic residue resulting from the UV irradiation of a H₂O:13CH₃OH:NH₃ (10:3.5:1 ratio) ice mixture by Meinert et al. (Science 2016).

Addressing these points could make the comparison between astrophysical ice analogs and meteorites more meaningful.

2. While the formation mechanism is unclear, the manuscript offers a few possibilities for the synthesis of deoxysugar derivatives. The manuscript also states that deoxysugar and their derivatives cannot form via mechanisms that involve the formose reaction. However, ice compositions here (H₂O:13CH₃OH) and those used by Meinert et al. (H₂O:13CH₃OH:NH₃) are fairly similar. It would be a reasonable assumption that a photochemically initiated formose reaction occurred in order to explain the diversity of sugars and derivatives (at least for the non-deoxy forms). Butscher et al has shown that VUV photolysis of formaldehyde in water-dominated ices results in the formation of glycolaldehyde, and only trace amounts of glycolaldehyde are needed to initiate a formose reaction. Furthermore, Snytnikova et al. has shown that photo-induced formose reactions do not require basic conditions or a catalyst (albeit these experiments were in aqueous solution).

Why are photo-induced formose reactions ruled out here (they are not discussed at all)? The typical photon energies of these experiments (~10.5 eV, 118 nm) are enough to photo-ionize and break bonds of typical sugar molecules (for example, see Ghosh et al. or Shin and Bernstein). Could this

ultimately lead to photo-induced deoxygenation of ribose and/or other sugars? Or modifications via radical interactions upon warming from intermediates? This will remain unanswered for now. However, a better comparison between the structural similarity of sugar derivatives (using the authors' previous finding in addition to the Meinert et al. paper) and deoxysugar derivatives (these results) may hint at possible formation mechanisms (i.e. for every sugar derivative, is there a deoxysugar derivative present as well? – Can this be addressed more in the paper?).

Interestingly, de Marcellus et al. (PNAS 2015) identified numerous aldehydes (C1-C4), including formaldehyde and glycolaldehyde (the first product in the formose reaction) in the organic residue resulting from the UV irradiation of a H₂O:13CH₃OH:NH₃ (12:3.5:1 ratio) ice mixture; however, when NH₃ was not present in the initial ice mixture (H₂O:13CH₃OH = 3:1), glycolaldehyde, glyceraldehyde (the second product in the formose reaction), and other aldehydes were not detected, which suggested that NH₃ played a vital role in the synthesis of these organic compounds. Taking NH₃ into account, it is quite possible that a formose-like reaction may not be responsible for the synthesis of deoxysugars in H₂O:CH₃OH ices. I wonder if the authors have any additional comments on this.

Finally, a quote in one of Eschenmoser's papers comes to mind, "since the outset of prebiotic chemistry, it has been taken for granted that sugars were primordially formed by way of the formose reaction. Convincing as this presumption may appear, it could be a mistake to conclude that there is neither room nor the need for looking out for alternatives." I would hope this group as well as others pursue the answers to these questions some more.

3. What is the reason for the range of abundances for deoxysugars in residues? For example, the abundance of 2-deoxyribose ranges from 217-3855 pmol. If these residues were performed under the same conditions and worked-up and analyzed in the same way, what is the reason for an order of magnitude difference? This can be seen for other deoxysugar derivatives in Table 1 as well.

Also in Table 1, it seems like the abundances of deoxysugars and derivatives are higher for higher molecular weight compounds. In Cooper et al., they state that a general decrease in abundance with increasing carbon number for sugar derivatives in meteorites was observed. What are the possible explanations for these differences?

4. I would note that Abreu and Bullock (2013) conducted an investigation of GRA 06100 and suggested that this meteorite has experienced higher temperatures (exceeding 600 °C) compared to other CR2 chondrites. I would expect that the diversity and abundance of soluble organic compounds in GRA 06100 to be much less compared to Murchison (the two meteorites that are compared to ice analogs here). Is this the case? This may also be a possible explanation as to why

few sugar derivatives were found in GRA 06100, and could be included in the discussion section. Also, Laino et al. have shown that 1,2-propanediol remains intact after pyrolysis at 800 K (which may help explain its presence in GRA 06100) and that 1,2-propanediol is stable at high temperature.

5. For Figure 2a, why is there a shift in the retention time (about 0.15 min) for 2-methyl-1,3-propanediol? These retention time shifts are not observed for any other unlabeled and labeled pairs of compounds in Figures 1-3. Also, the mass spectrum for 2-methyl-1,3-propanediol reference standard shows a completely different ratio for m/z 261 and 133 compared to the compound measured in the ice residue. Additionally, there is another large m/z peak in the ice residue not found in the standard spectrum. What is the reason for this? Are there interfering compounds present (do they show up in the TIC)? How can a conclusive identification be made in this case? Finally, what is the chromatographic retention time of the 2-methyl-1,3-propanediol standard; how well does it compare with the compound in the ice residue (this is not shown). A brief explanation is warranted.

Reviewer #3 (Remarks to the Author):

Nuevo et al. present in this manuscript experimental data on the formation of sugar related compounds in residues produced by photochemistry in ice. They have exposed ice mixture of water and methanol to UV irradiation and detected, in their residue, several deoxysugar, deoxysugar alcohols and deoxysugar acids among other products. Of particular interest, 2-deoxyribose, the sugar constituent of DNA, is present in the UV-irradiation products. Of note, the authors conducted some of the experiments using ^{13}C -labelled methanol in order to rule out contamination in their experiment. They also looked for the same compounds in Murchison and GRA 06100.

Although I believe that the authors present here is a robust study and that the interpretations drawn by the authors are consistent, this manuscript is not suitable for the broad audience of Nature Communications. I would rather recommend this manuscript to be submitted to a more specialized journal in chemistry or astrophysics. First of all, the manuscript is very focused on chemistry and astrobiology, see for instance the figures, all being chromatograms and mass spectra. Secondly, the manuscript does not contain significantly novel results or conclusions that would fit with the editorial recommendations of Nature Communications. Indeed, the important discovery that sugar and their derivatives can be synthesized by UV irradiation of ices has been already published elsewhere (Meinert et al. in 2016, cited by the authors). The main point of the authors here is that they identified 2-deoxyribose while Meinert et al. did not; although Meinert et al. reported ribose. I do not see where the novelty is; finding deoxysugars on top of sugars in ice experiments does not constitute, to my sense, a major breakthrough. A chemical mechanism for the formation of all the

sugars in ice experiments would be new, but unfortunately, as the authors declare at lines 159-161, the mechanism remains unclear.

I'm not convinced that the authors can show that these sugars could have been delivered to Earth. The two studied chondrites do not contain the same sugars (or deoxysugars) as the one formed in ice experiments. This is interpreted as the low stability of these compounds compared to their alcohol or acid derivative; I agree with that. Then, there is a glitch in the reasoning of the paper. If sugars and deoxysugars were synthesized in the ice grains that were accreted onto the chondrite parent bodies, secondary processes would have destroyed them, hence preventing the delivery of these sugars and deoxysugar at the surface of the primitive Earth. The conclusion drawn at lines 217-219 is then wrong. Or the authors have a mechanism to form sugars from their more stable alcohol or acid derivatives on the primitive Earth. This would have to be supported by experiments, not performed here, or by references to other studies. I do not think that DNA or RNA could form with sugars not being in their aldose forms.

My main last concern with this manuscript is the connection made with the old scenario that building blocks of life on Earth were brought from space. This is not new and the literature about extraterrestrial organic matter is full of similar statements. In the end, this statement is really overrated. The main criticism one may argue to this hypothesis is the concentration of these organic compounds in terrestrial prebiotic lakes or ocean. Even considering that 100% of organic compounds would survive the fall on Earth, the amount delivered would be too small to achieve a concentration high enough to induce chemical reactions similar to biochemical ones. And if we suppose repetitive falls of comets or meteorites at a specific location, to increase the quantity of delivered organic matter, then the energy resulting from these impacts would actually hamper the lifetime of organic molecules. The concern of the concentration of extraterrestrial organic compounds at the surface of the prebiotic Earth is never really discussed in the exobiology literature and this manuscript does not really care about it. For me, making the direct conclusion that, because these sugars could be produced by some processes in space, the building blocks of the first DNA molecules on Earth (or RNA and other biomolecules) are of extraterrestrial origin is the main weakness of the rationale of this study (see the beginning of the abstract).

A few minor comments:

At line 84 it is said that "All the residues produced in this work contained a wide variety of sugars, sugar alcohols, and sugar acids, with a similar distribution to previous laboratory studies" in addition to which they detected deoxysugar derivatives reported in table 1. I recommend to report the concentration of all the sugar related compounds, including compounds already reported in previous studies. That would be useful for comparison with future experiments.

At line 115: it is said that yields are very variable and later it is said that the abundance of products in ^{13}C -labelled experiments is lower. Why is that? How reliable are the yields reported here if there is so much variation?

As a summary of my review, this is a robust experimental study that should be submitted elsewhere, in a more specialized journal. The authors could then describe into more details the processes occurring in their ice residue, discuss (potentially new?) mechanisms for formation of complex molecules in space and soften a lot the connection between their data and the influence of extraterrestrial organic matter on the emergence of life.

Reviewer #1 (Remarks to the Author):

The manuscript entitled ‘Deoxyribose and deoxysugar derivatives in residues produced from photoprocessed astrophysical ice analogues and comparison to meteorites’ deals with the formation of 2-deoxyribose and other deoxysugar derivatives in residues by ultraviolet irradiation of ice mixtures consisting of H₂O and CH₃OH.

Overall this is a very interesting study of broad interest.

Thank you, we tried to make this study available to the broadest readership possible.

Here are my comments:

1. I recommend to remove the statement, that the ‘organics that triggered life are of extraterrestrial origin’ in the second sentence of the abstract paragraph. The question would be, why only on extraterrestrial objects and not on earth itself. There has been a recent publication about the formation of sugars triggered by Schreibersite, which is commonly found in meteorites Pallmann, S. et al. Schreibersite: an effective catalyst in the formose reaction network. *New J. Phys.* 20, 055003 (2018). This is a plausible mechanism, that sugars are formed on earth. It would be very good to provide a high-resolution MS/MS spectrum of the potentially formed deoxyribose for absolute confirmation.

We thank the Reviewer for bringing this paper to our attention, which shows that formose-type reactions (hereafter, FTR) could have contributed to the formation of sugar and sugar derivatives catalyzed by schreibersite via aqueous alteration in meteorite parent bodies (asteroids). However, this does not necessarily imply that such reactions have taken place on Earth. Schreibersite is found in iron-rich meteorites (e.g., Magura, Sikhote-Alin, São Julião de Moreira, Gebel Kamil), and very rarely on Earth, the only exception being the Disko Island, Greenland (Bryant et al. 2013, *Geochim. Cosmochim. Acta*, **109**, 90). These meteorites typically contain little to no organics, and therefore may not contain high amounts of sugar derivatives. The idea that terrestrial schreibersite may have played a role in the origin of life, but its presence in meteorites containing little organics and only one place on Earth limits the range of its role. There is nonetheless the possibility that some schreibersite may have been brought to the surface of the primitive Earth via bombardment of iron-rich, organic-poor meteorites, which could have interacted with the organics delivered via organic-rich meteorites that fell nearby. However, this would require a very intense bombardment in which a large number of meteorites of each type fall within an area small enough for these materials to interact. We are not aware of such a study, so it is difficult to evaluate its feasibility.

In any event, we have modified the corresponding sentences in the abstract, introduction, and conclusion to make it clear that while the delivery of extraterrestrial organics may have played a role in the origin of life, planetary processes must also have been critical.

Finally, we are not sure whether we understand the Reviewer’s comment about “providing a high-resolution MS/MS spectrum of the potentially formed deoxyribose”, so it is difficult to address it. If the Reviewer meant that high-resolution mass spectrometry may help distinguishing

between organic compounds of similar molecular masses, it still would not help distinguishing between isomers of exactly the mass, which is the case for diastereoisomers of sugar derivatives such as deoxyribose and deoxyxylose.

2. Is there a chance to give more insights in the formation mechanism? Methanol has to be oxidized in the reaction. How does this work. Is there residual oxygen or is it a dehydration reaction that is triggered by a surface reaction under UV irradiation. It is interesting, that apparently the selectivity to form sugars starting from methanol is better than starting with formaldehyde. The question is also, if formaldehyde polymerizes under these conditions or if it is still molecular formaldehyde?

The question of the formation mechanism(s) is important, however, it cannot be answered easily, because the processes taking place in the ices during such UV irradiation experiments are complex and stochastic in nature, as the chemistry is dominated by reactions between radicals and ions, at temperatures that significantly limit the mobility of the reacting species.

In the manuscript, we added several sentences in the Discussion section to briefly discuss this point by focusing mainly on the stochastic nature of the processes and the fact that there are probably *several* mechanisms leading to the formation of sugar derivatives rather than one unique pathway. We also focus on the formation of deoxysugar derivatives rather than all sugar derivatives. In the text in green font below, the Reviewer will find a more detailed description of such chemical processes, showing why formation mechanisms cannot be addressed in detail in this manuscript, in case he/she wants a more detailed answer to this question. However, since most of the details given below are not covered in the current manuscript, the Reviewer is free to skip this detailed explanation.

Finally, to answer the Reviewer's question regarding the polymerization of formaldehyde into polyoxymethylene (POM), the presence of POM in photo-irradiated ices has been reported in several previous studies (e.g., Schutte et al. 1993, *Science*, **259**, 1143; Bernstein et al. 1995, *Astrophys. J.*, **454**, 327; Bernstein et al. 1997, *Adv. Space Res.*, **19**, 991; Muñoz Caro & Schutte 2003, *Astron. Astrophys.*, **412**, 121). Formaldehyde being a well known photoproduct of methanol, the presence of POM in our residues is expected, together with formaldehyde monomers. Note, however, that if a non-negligible fraction of the formaldehyde is locked-up in its POM form, it cannot be involved easily in the formation of sugar derivatives via a formose-type reaction (see below if interested).

Further details associated with the difficulty in establishing the formation mechanisms.

In the following, we want to provide a detailed explanation regarding why the question of the formation mechanisms of sugar (and deoxysugar) derivatives cannot be fully elucidated in this manuscript. Chemical reactions in ice mixtures induced by UV radiation and taking place at very low temperatures (< 20 K) are very different from what occurs in classical chemistry driven by thermodynamics. Indeed, the absorption of UV photons by H₂O and CH₃OH leads to the creation of ions, radicals, and species in electronically excited states that will react with their neighbours. Such reactions do not necessarily lead to most stable products possible, however. At such low temperatures, ions, radicals, and excited species have very low mobility and thus mostly react with their closest neighbours. In this chemical framework, the distribution of photoproducts does not follow a thermodynamic distribution, but rather a *stochastic* distribution in which the abundances of the photoproducts typically decrease exponentially with the length of the carbon chain. This is

because the formation of larger products requires a higher number of photon absorptions—and therefore reaction steps—to take place.

In addition, it must be noted that photons absorbed by the ice can also lead to the *destruction* of compounds, so that the formation of a photoproduct is not a linear process, but rather the result of multiple formation and destruction reactions. Consequently, in such processes, the larger the molecule, the higher number of photons is required, and the number of photons is *not* linearly proportional to the number of steps required for the formation of a given compound.

An example that the UV irradiation of ice mixtures does not follow single, established mechanistic pathways is described in Elsilá et al. (2007, *Astrophys. J.*, **660**, 911). In that study, the authors describe an attempt to determine the formation mechanism(s) of amino acids under experimental conditions similar to ours. They used isotopically labelled molecules in their starting ices so they could follow how these were incorporated into the amino acids that were produced. The authors conclude that there is not *one* mechanism leading to the formation of the same amino acids, but *several* of them. Following this reasoning, it becomes clear that determining the formation mechanism(s) for sugar derivatives and other organic compounds under our experimental conditions is a very complex task, as any photoproduct likely forms via a *multitude* of different pathways, some of which may involve the formation of even large molecules that are subsequently broken down.

This being said, what we do know is that the formation of sugar derivatives must come from the methanol, because it is the only carbon source in the starting ice mixtures. When methanol absorbs a UV photon, it can be ionized, or one of its molecular bonds can be cleaved off to form $\bullet\text{CH}_2\text{OH}$, $\text{CH}_3\text{O}\bullet$, and $\bullet\text{CH}_3+\bullet\text{OH}$, radicals, as well as ionized variants of these radicals. These very reactive species may recombine to either form methanol again or form new, larger molecules, which can then themselves react with other radicals or create new radicals that will react with other species around them, and so on. Öberg et al. (2009, *Astron. Astrophys.*, **504**, 891) showed that the branching ratios for the radicals created from the UV photolysis of methanol are $\bullet\text{CH}_2\text{OH}:\text{CH}_3\text{O}\bullet:\bullet\text{CH}_3+\bullet\text{OH} \approx 5:1:1$, suggesting that $\bullet\text{CH}_2\text{OH}$ radicals are dominant in the ice matrix. The recombination of two of these radicals may lead to the formation of ethyleneglycol (see, e.g., Öberg et al. 2009; Maity et al. 2015, *Phys. Chem. Chem. Phys.*, **17**, 3081; Butscher et al. 2016, *Astron. Astrophys.*, **593**, A60), while the recombination of three of them may lead to the formation of glycerol (after the loss of one H atom). Glycerol is by far the most abundant sugar derivative formed when ice mixtures containing CH_3OH are UV irradiated, as observed by Meinert et al. (2016) as well as in experiments performed in our laboratory (unpublished results).

The role of H_2O in the ice matrix is usually multiple: (1) provide $\bullet\text{OH}$ radicals via $\text{H}_2\text{O} \rightarrow \bullet\text{OH} + \bullet\text{H}$; (2) play the role of a third body in the formation of photoproducts, by absorbing the excess energy of intermediate species towards the formation of stable photoproducts, as it was studied in detail experimentally and theoretically for the formation of pyrimidine-based nucleobases from the UV irradiation of pyrimidine in H_2O -rich ices (see, e.g., Nuevo et al. 2009, *Astrobiology*, **9**, 683; Bera et al. 2010, *J. Chem. Phys.*, **133**, 104303; Materese et al. 2013, *Astrobiology*, **13**, 948; Sandford et al. 2015, *Photoinduced Phenomena in Nucleic Acids II, Topics in Current Chemistry*, **356**, 123; Bera et al. 2016, *J. Chem. Phys.*, **144**, 144308); and (3) shield newly formed molecules against incoming UV photons and decrease their probability of being destroyed.

Another mechanism to form sugar derivatives, mentioned by the Reviewer and suggested by Meinert et al. (2016), is the formose reaction (hereafter, FR), or more generally formose-type reactions (hereafter, FTR), involving the oligomerization of formaldehyde (H_2CO). Formaldehyde

is a well-known photoproduct of UV-irradiated methanol ice, and it is therefore expected to be present in the ice matrix. However, the formation of formaldehyde itself from methanol, the only source of carbon in our starting ice mixtures and in the ice mixture in the experiment described in Meinert et al. (2016), requires several reaction steps, so that this pathway has high chances to be rerouted towards the formation of species *other* than formaldehyde. Nevertheless, the formaldehyde that does form can oligomerize via FTR and lead to the formation of a fraction of the sugar derivatives detected in the residues. How big this fraction coming from FTR is, however, nearly impossible to quantify, given the huge number of possible reactions taking place in these experiments.

However, an FTR mechanism has its own caveats, as such reactions usually take place in an aqueous solution (i.e., liquid phase), at temperatures of 50°C or higher, i.e., much higher than 15 K (our study) or 80 K (Meinert et al. 2016's study), in the presence of a basic catalyst (typically Ca^{2+} based), and in the presence of one or several co-catalysts, also called inductors (e.g., glycolaldehyde, glyceraldehyde, dihydroxyacetone, or other larger sugars) in order to ignite the initial aldol condensation between two formaldehyde molecules, which is the limiting step of any FTR. The presence of the co-catalysts in the starting mixtures helps speeding up this first reaction step and trigger the autocatalytic FR towards the formation of larger sugars and sugar derivatives (see, e.g., Delidovich et al. 2014, *ChemSusChem*, 7, 1833). These conditions are however quite different from the UV irradiation experiments of ice mixtures at low temperature described here and in the Meinert et al. (2016) paper. The first, limiting step of the FR may be initiated by UV irradiation, as suggested by Meinert et al. (2016) and other studies (e.g., Delidovich et al. 2014; Snytnikova et al. 2006, *Mendeleev Commun.*, 9), in particular to help the formation of small 2C and 3C co-catalysts and trigger the FR.

In any case, the FR as formation mechanism for sugar derivatives from the UV irradiation of ices was proposed by Meinert et al. (2016) as pure *speculation* and their study showed *no experimental proof* of such a mechanism. We could have done the same and added a few sentences in our manuscript saying that the deoxysugar derivatives produced in our residues are formed via photo-deoxygenation of the corresponding sugar derivatives, which themselves are formed via FTR, with no experimental proof. But because of the stochastic nature of the processes taking place in such experiments, we know that this is not entirely true. One indication that the FR mechanism is not likely to be the *only* mechanism involved in these reactions is that, as shown in the Fig. 3 of the Meinert et al. (2016) paper, the FR mostly leads to the formation of sugars, and then the sugar alcohols and sugar acids are formed via reduction or oxidation of the sugars. No matter what these reduction and oxidation mechanisms are (e.g., Cannizzaro-type reactions), this means that the formation of sugar alcohols and sugar acids require *at least one additional reaction step* from the corresponding sugars, i.e., one or more additional photons need to be absorbed to reduce/oxidize a given sugar, *without destroying it*. In addition, the distribution of sugar derivatives in the unique residue studied by Meinert et al. (2016) as well as in several (>20) residues studied in our laboratory (unpublished results) shows that the *sugar alcohols* are always more abundant than their corresponding sugars. This result rather favours a mechanism involving the recombination (oligomerization) of methanol and/or $\bullet\text{CH}_2\text{OH}$ radical units, as we suggested above. Sugars are the second most abundant category of sugar derivatives detected in the residue analysed by Meinert et al. (2016) (see their Fig. 1) and in our residues (unpublished), and then finally sugar acids (least abundant). This shows a trend in which sugar alcohols (least oxidized sugar derivatives) were formed before their corresponding sugars and then sugar acids. Again, this

result does *not* exclude the possibility that some of the sugars and sugar derivatives may be formed via FTR, but it clearly tells us that FTR are probably not the only mechanisms involved.

Regarding the formation of *deoxysugar* derivatives, the wide range spanned by the abundance ratios between deoxyribose and ribose in our residues supports the idea that these compounds may form via a multitude of distinct pathways. For instance, some of the pathways leading to the formation of deoxyribose may include the recombination of smaller intermediate species, with one lacking an OH group compared with those recombining to form ribose, while other pathways may involve the formation of ribose first, via any combination of the mechanisms discussed above, followed by the loss of an OH group. Therefore, determining formation mechanism(s) for such compounds requires a full new, long series of experiments that would be fully dedicated to answering this question, which, although very interesting, is not possible to do within a reasonable timeframe for this current paper. This important topic may thus be discussed in detail in a future paper.

3. All experiments performed are well described.

Thank you, our experiments and analyses were performed carefully and the results presented are based on the analysis of several samples, as opposed to the study from Meinert et al. (2016), who discussed results obtained for only *one* residue.

Reviewer #2 (Remarks to the Author):

Review of NCOMMS-18-16476

Deoxyribose and deoxysugar derivatives in residues produced from photoprocessed astrophysical ice analogues and comparison to meteorites

This manuscript reports the detection of 2-deoxyribose and several deoxysugar derivatives in room-temperature residues produced from the VUV-irradiation of ice mixtures containing water and methanol (or ^{13}C -labeled methanol). This manuscript is well written, the data is clearly presented, and the identification of deoxysugar compounds appear sound (mass spectra comparison to reference standards is convincing and the use of isotopic labels helps rule out the possibility that products are the result of contamination). The detection of 2-deoxyribose (along with the previous detection of ribose by Meinert et al.) in these reactions is highly significant since these organic compounds are critical to the structure of DNA (and RNA) and would have important implications to the fields of cosmochemistry and origin of life/astrobiology. This is an important paper and should be published in Nature Communications.

We thank the Reviewer for his/her positive comments and recommendation for our work to be published in *Nature Communications*.

I do have a few minor points that I hope the authors could address:

1a. Deoxysugar acids appear to be diverse in carbonaceous chondrites in Cooper et al. (Nature 2001). Yet only one deoxysugar acid (3,4-dihydroxybutyric acid; Table 1) was identified in residues of VUV-irradiated H_2O^{12} or $^{13}\text{CH}_3\text{OH}$ ice mixtures in this report. *Were other deoxysugar acids (4C, 5C, and 6C) searched for in these residues?* 2-Methylglyceric acid, 2,4-dihydroxybutyric acid, 2,3-dihydroxybutyric acid, 2-deoxypentonic acids, 2-deoxyhexonic acids, and 3-deoxyhexonic acids were previously identified in Murchison and Murray meteorites by Cooper et al. If these compounds were searched for and not found in astrophysical ice analogs, what could be some possible explanations for the absence of these deoxysugar acids?

Yes, we did search for 4C and 5C deoxysugar acids in these residues, but they were not detected. This could mean that they are not present, that they have abundances too small for us to detect, or that the comparison of their peaks in the sample chromatograms with chromatograms of the relevant standards was inconclusive. After our manuscript was first submitted to *Nature Communications*, we purchased and ran four additional deoxysugar acid compounds, namely, the four isomers of 2,3-dihydroxybutyric acid (L and D enantiomers of the *threo* and *erythro* isomers, which elute around 27 and 32 min), but could not find any of them in the residues. However, as mentioned in the manuscript, there is a pair of peaks eluting around 74.3 and 74.7 min whose mass spectra are identical and are consistent with the presence of both enantiomers of a 4C deoxysugar acid. One possibility is 3,4-dihydroxybutyric acid, for which we did not have the standard, but for which the mass spectrum of each of these peaks matches that of the NIST database standard for such a compound (see Fig. S4). The fact that these are high retention times may however be due to the presence of the dimer of such a 4C deoxysugar acid rather than the monomer, which is expected to elute at shorter retention times, like the four 2,3-dihydroxybutyric acid isomers do. In

addition, the distribution of the sugar derivatives found in carbonaceous chondrites (Cooper et al. 2001) does *not* match that of the sugar derivatives identified in laboratory residues produced from the UV irradiation of CH₃OH-rich ices (Meinert et al. 2016; unpublished results from our laboratory). Indeed, with the exception of dihydroxyacetone, no sugars were detected in meteorites, while in laboratory residues sugars with up to 5 carbon atoms have been identified. Also, in meteorites, sugar alcohols and sugar acids are present in a wide variety, with glycerol (sugar alcohol) and glyceric acid (sugar acid) being very abundant, while in laboratory residues sugar alcohols are significantly more abundant than their corresponding sugar acids (see Fig. 1 of Meinert et al. 2016, with the exception of glycerol and glyceric acid, which have similar abundances). The same trend may also be true for deoxysugar derivatives, i.e., deoxysugar acids may be much more abundant in meteorites than in laboratory residues, while deoxysugars such as 2-deoxyribose may not be detected in meteorites while some of them are in laboratory residues (this study). The differences in distribution between meteorites and laboratory residues may have several causes. Indeed, these differences may come from the fact that in the laboratory experiments described in our study and in Meinert et al. (2016), methanol is the only source of carbon in the starting ice mixtures. The addition of other carbon sources known to be present in astrophysical ices such as CO, CO₂, CH₄, etc., may have an effect on the distribution of products in the final residues. Another cause for such discrepancies may be due to the fact that some meteorite parent bodies may have experienced aqueous alteration, which has not been studied in detail to date. Future studies will cover these two hypotheses.

1b. It is interesting to note that many of the sugar alcohols and sugar acids (3C, 4C, and 5C) in Murchison and Murray meteorites identified by Cooper et al. were also identified in the organic residue resulting from the UV irradiation of a H₂O:¹³CH₃OH:NH₃ (10:3.5:1 ratio) ice mixture by Meinert et al. (Science 2016).

This is correct, and we are currently analysing more than 20 residues produced from different ice mixtures (with different combinations of carbon sources) at 12 K. Results show that residues produced from mixtures containing only CH₃OH as the carbon source, together with H₂O and in the presence or absence of NH₃, show very similar results to those reported by Meinert et al. (2016), i.e., they contain sugar alcohols and sugar acids, some of which are similar to those found in Murchison and Murray. However, such residues also contain sugars in higher quantities compared to sugar acids, whereas no sugars were found in meteorites, with the exception of the ketose sugar dihydroxyacetone.

1c. Addressing these points could make the comparison between astrophysical ice analogs and meteorites more meaningful.

As we mentioned in our two previous answers, these comparisons between the distribution of sugar derivatives in laboratory residues and in meteorites will be covered in more detail in an upcoming paper. For this current manuscript, we only focus on the identification and discussion regarding deoxysugar derivatives.

2a. While the formation mechanism is unclear, the manuscript offers a few possibilities for the synthesis of deoxysugar derivatives. The manuscript also states that deoxysugar and their derivatives cannot form via mechanisms that involve the formose reaction. However, ice

compositions here ($\text{H}_2\text{O}:\text{}^{13}\text{CH}_3\text{OH}$) and those used by Meinert et al. ($\text{H}_2\text{O}:\text{}^{13}\text{CH}_3\text{OH}:\text{NH}_3$) are fairly similar. It would be a reasonable assumption that a photochemically initiated formose reaction occurred in order to explain the diversity of sugars and derivatives (at least for the non-deoxy forms). Butscher et al. has shown that VUV photolysis of formaldehyde in water-dominated ices results in the formation of glycolaldehyde, and only trace amounts of glycolaldehyde are needed to initiate a formose reaction. Furthermore, Snytnikova et al. has shown that photo-induced formose reactions do not require basic conditions or a catalyst (albeit these experiments were in aqueous solution).

The answer to this comment as well as the modifications made to the manuscript regarding this particular point have been covered in detail in the answer to Reviewer #1's comment 2.

2b. Why are photo-induced formose reactions ruled out here (they are not discussed at all)? The typical photon energies of these experiments (~ 10.5 eV, 118 nm) are enough to photo-ionize and break bonds of typical sugar molecules (for example, see Ghosh et al. or Shin and Bernstein). Could this ultimately lead to photo-induced deoxygenation of ribose and/or other sugars? Or modifications via radical interactions upon warming from intermediates? This will remain unanswered for now. However, a better comparison between the structural similarity of sugar derivatives (using the authors' previous finding in addition to the Meinert et al. paper) and deoxysugar derivatives (these results) may hint at possible formation mechanisms (*i.e. for every sugar derivative, is there a deoxysugar derivative present as well? – Can this be addressed more in the paper?*).

We could not find the papers the Reviewer referred to in his/her comment, for lack of information about these references (journal, year, etc.). In any case, a detailed answer to the question of the formation of sugar derivatives via a formose-type reaction process as well as the possible mechanism(s) for the formation of deoxyribose can be found in the answer to Reviewer #1's comment 2.

2c. Interestingly, de Marcellus et al. (PNAS 2015) identified numerous aldehydes (C1-C4), including formaldehyde and glycolaldehyde (the first product in the formose reaction) in the organic residue resulting from the UV irradiation of a $\text{H}_2\text{O}:\text{}^{13}\text{CH}_3\text{OH}:\text{NH}_3$ (12:3.5:1 ratio) ice mixture; however, when NH_3 was not present in the initial ice mixture ($\text{H}_2\text{O}:\text{}^{13}\text{CH}_3\text{OH} = 3:1$), glycolaldehyde, glyceraldehyde (the second product in the formose reaction), and other aldehydes were not detected, which suggested that NH_3 played a vital role in the synthesis of these organic compounds. Taking NH_3 into account, it is quite possible that a formose-like reaction may not be responsible for the synthesis of deoxysugars in $\text{H}_2\text{O}:\text{CH}_3\text{OH}$ ices. I wonder if the authors have any additional comments on this.

The vacuum system used to perform the experiments described in our manuscript has never seen any NH_3 (nor other N-bearing compounds), and we will keep it free of NH_3 , because NH_3 is a background contaminant which is extremely difficult to get rid off. For this reason, we are sure that the photoproducts detected in our residues are in no way associated with the presence of NH_3 , at any stage of the process. This being said, the fact that deoxysugar derivatives (and sugar derivatives, unpublished results) are detected in all residues, regardless of the presence (Meinert et al. 2016) or absence (our work) of NH_3 in the starting ice mixture, indicates that (i) NH_3 is not

required to form these compounds, and (ii) the formose reaction is probably not the only formation mechanism involved in these processes.

2d. Finally, a quote in one of Eschenmoser's papers comes to mind, "since the outset of prebiotic chemistry, it has been taken for granted that sugars were primordially formed by way of the formose reaction. Convincing as this presumption may appear, it could be a mistake to conclude that there is neither room nor the need for looking out for alternatives." I would hope this group as well as others pursue the answers to these questions some more.

We agree with this citation on the fact that the formose reaction is not the universal answer to the question of the formation of sugars and their derivatives, but only one possibility among others, as supported by our results.

3a. *What is the reason for the range of abundances for deoxysugars in residues?* For example, the abundance of 2-deoxyribose ranges from 217-3855 pmol. If these residues were performed under the same conditions and worked-up and analyzed in the same way, what is the reason for an order of magnitude difference? This can be seen for other deoxysugar derivatives in Table 1 as well.

The composition of the starting ices in all our experiments is the same within the uncertainty of the partial pressures of H₂O and CH₃OH in the gas mixtures when preparing the gas mixtures in the bulbs. Similarly, we do everything we can to ensure consistent temperatures and radiation doses during our experiments. Nonetheless, there may be slight variations in the conditions that can lead to different absolute and relative abundances of photoproducts in the final residues. This is due to the stochastic nature of the processes taking place during these irradiations (see our detailed response to Reviewer #1's comment 2). This has been observed previously in many experiments of UV irradiation of ices for the formation of amino acids (e.g., Bernstein et al. 2002, *Nature*, **416**, 401; Munoz Caro et al. 2002, *Nature*, **416**, 403; Nuevo et al. 2006, *Astron. Astrophys.*, **457**, 741; Nuevo et al. 2007, *Adv. Space Res.*, **39**, 400; Nuevo et al. 2008, *Orig. Life Evol. Biosph.*, **39**, 38). Differences may also appear during the warm-up phase, during which ices are more mobile and their structural composition changes. For example, when H₂O ice changes from one crystalline phase to another, some of the volatile molecules, including small photoproducts, may sublime away, and this sometimes results in the desorption of larger molecules in the process, resulting in a net loss of photoproducts that are released to the gas phase and do not re-condense on the substrate. Therefore, the photoproducts detected may end up having different absolute and relative abundances from one residue to another, but the important thing to remember is that *the same variety of organics is always produced* when similar ice mixtures are UV irradiated under similar experimental conditions. The reason why the Meinert et al. (2016) paper does not discuss this point is simply because *they only reported the results obtained for one residue*, so there is no information regarding the reproducibility of their experiment, or how the distribution of photoproducts may vary from one residue to another.

3b. Also in Table 1, it seems like the abundances of deoxysugars and derivatives are higher for higher molecular weight compounds. In Cooper et al., they state that a general decrease in abundance with increasing carbon number for sugar derivatives in meteorites was observed. What are the possible explanations for these differences?

Due to the stochastic nature of the chemistry taking place during these experiments (see our detailed answer to Reviewer #1's comment 2), the abundances of the organic compounds produced are expected to decrease with the length of their carbon chain. Such a distribution is globally observed for regular sugars, sugar alcohols, and sugar acids containing 3 to 6 carbon atoms in our residues (unpublished results) as well as in Meinert et al. (2016), with the exception that they did not detect erythrose and threose (4C sugars), while they detect ribose and its isomers (5C sugars) as well as 4C–5C sugar alcohols and sugar acids. We do detect 4C sugars in all our residues produced from H₂O:CH₃OH ice mixtures (unpublished results). Regarding smaller (2C) compounds and 3C–4C deoxysugar derivatives, it is not clear what happens, however, we can propose a couple of explanations regarding their low abundances in the final residues. First, smaller compounds tend to sublime away more easily during the warm-up stage, as ices are evaporating, because they are not stable in the solid phase at higher temperatures. As a result, they are under-represented in the final room temperature residues. However, some of these volatile compounds may also be trapped within the refractory residue that is recovered at room temperature, which is why we can still detect them in small abundances. A second reason why smaller deoxysugar compounds cannot be easily detected with our analytical method is because they contain a too small number of OH groups in their structure, so that they may still be too volatile to be seen in the GC-MS chromatograms even when derivatized, because they evaporate before or together with the solvent. The main GC-MS method used in this work ((+)-2-butanol/TFAA) does not allow for the detection of such small, volatile compounds which contain fewer than 3 OH groups (which is the case for 3C and 4C deoxysugars). Therefore, for a few samples we used an alternative method (BSTFA) to search for these smaller compounds. Among those smaller compounds, only 1,2- and 1,3-propanediol could be detected in low abundances, perhaps because they were bound to other compounds in the residue, preventing them from fully subliming away during the warm-up stage.

In the Supplementary Information, we added a sentence to mention that the GC-MS method we used does not allow for the detection of smaller deoxysugar derivatives. In the particular case of 1,2-propanediol, it could also be because of what the Reviewer mentions in the next comment about its stability at high temperatures. In this case, its low abundance could be explained by a loss during the warm-up process. Another sentence was added to the manuscript regarding this particular point, together with the Abreu & Bullock (2013) and Laino et al. (2012) references (see next answer).

4. I would note that Abreu and Bullock (2013) conducted an investigation of GRA 06100 and suggested that this meteorite has experienced higher temperatures (exceeding 600 °C) compared to other CR2 chondrites. I would expect that the diversity and abundance of soluble organic compounds in GRA 06100 to be much less compared to Murchison (the two meteorites that are compared to ice analogs here). Is this the case? This may also be a possible explanation as to why few sugar derivatives were found in GRA 06100, and could be included in the discussion section. Also, Laino et al. have shown that 1,2-propanediol remains intact after pyrolysis at 800 K (which may help explain its presence in GRA 06100) and that 1,2-propanediol is stable at high temperature.

We thank again the Reviewer for bringing these two interesting papers to our attention. 1,2-Propanediol is the only compound we could detect in the GRA 06100 sample we analysed, and this result is consistent with the fact that this compound seems to be stable at high temperature, even as high as those GRA 06100's parent body experienced. We added these facts to the

manuscript and added the two references cited by the Reviewer. Consequently, we also deleted and replaced a few other references in the text and the list so that the total number of references does not exceed 50.

5. For Figure 2a, why is there a shift in the retention time (about 0.15 min) for 2-methyl-1,3-propanediol? These retention time shifts are not observed for any other unlabeled and labeled pairs of compounds in Figures 1-3. Also, the mass spectrum for 2-methyl-1,3-propanediol reference standard shows a completely different ratio for m/z 261 and 133 compared to the compound measured in the ice residue. Additionally, there is another large m/z peak in the ice residue not found in the standard spectrum. What is the reason for this? Are there interfering compounds present (do they show up in the TIC)? *How can a conclusive identification be made in this case?* Finally, what is the chromatographic retention time of the 2-methyl-1,3-propanediol standard; how well does it compare with the compound in the ice residue (this is not shown). A brief explanation is warranted.

It is not uncommon to see minor variations in retention times of this magnitude. Residues and standards were not always injected the same day, and in a few rare cases, several weeks or months apart. Because the GC-MS instrument is used for several different projects, after some time, retention times can shift a little. Also, the retention times can vary from one sample to another because of the interaction with other compounds in the residues which elute at similar times. One or both of these factors contributed to the shift seen between the peaks for 2-methyl-1,3-propanediol in Figure 2a. The reason why the intensity of the m/z 133 and 261 peaks do not match well in the mass spectra of 2-methyl-1,3-propanediol is because it is probably coeluting with another compound in the residues, and although it is usually possible to subtract the contribution from other near peaks with the GC-MS analysis software, in this case it was not possible.

To take this remark into account, we have added text to the manuscript and the captions of Table 1 and Figure 2 to say that 2-methyl-1,3-propanediol probably elutes with another unidentified compound, so that the abundances derived for this compound are only upper limits. Finally, the retention time of the 2-methyl-1,3-propanediol standard is 37.26 min, and it was injected several months after the residues shown in Figure 2a, because we were originally not searching for smaller compounds, and we injected some of the standards at a later time. This can explain the shifts in retention times observed here.

Reviewer #3 (Remarks to the Author):

Nuevo et al. present in this manuscript experimental data on the formation of sugar related compounds in residues produced by photochemistry in ice. They have exposed ice mixture of water and methanol to UV irradiation and detected, in their residue, several deoxysugar, deoxysugar alcohols and deoxysugar acids among other products. Of particular interest, 2-deoxyribose, the sugar constituent of DNA, is present in the UV-irradiation products. Of note, the authors conducted some of the experiments using ^{13}C -labelled methanol in order to rule out contamination in their experiment. They also looked for the same compounds in Murchison and GRA 06100.

1. Although I believe that the authors present here is a robust study and that the interpretations drawn by the authors are consistent, this manuscript is not suitable for the broad audience of *Nature Communications*. I would rather recommend this manuscript to be submitted to a more specialized journal in chemistry or astrophysics. First of all, the manuscript is very focused on chemistry and astrobiology, see for instance the figures, all being chromatograms and mass spectra. Secondly, the manuscript does not contain significantly novel results or conclusions that would fit with the editorial recommendations of *Nature Communications*. Indeed, the important discovery that sugar and their derivatives can be synthesized by UV irradiation of ices has been already published elsewhere (Meinert et al. in 2016, cited by the authors). The main point of the authors here is that they identified 2-deoxyribose while Meinert et al. did not; although Meinert et al. reported ribose. I do not see where the novelty is; finding deoxysugars on top of sugars in ice experiments does not constitute, to my sense, a major breakthrough. A chemical mechanism for the formation of all the sugars in ice experiments would be new, but unfortunately, as the authors declare at lines 159–161, the mechanism remains unclear.

We thank the Reviewer for mentioning that our study and the interpretation of its results in our manuscript were robust and consistent. Short scientific papers cannot always target the broadest audience possible, as it is the case in journals such as *Nature Communications* and other *Nature* journals. Thousands of papers are published in such journals and describe very specific research restricted to specific field, but this does not mean they are not targeting a broader readership. One example is the Cooper et al. (2001) paper, published in *Nature* and cited several times in our manuscript, in which the figures were all chromatograms and mass spectra of meteoritic samples, but whose results has an impact for a broad scientific community. We believe our manuscript also targets a broad audience among the scientific community, because our work is related astrobiology and to the origin of life. Reviewers #1 and #2 seem to agree with us.

Regarding the formation mechanism(s), we invite the Reviewer to read our answer to Reviewer #1's comment 2, where we address this question in detail.

2. I'm not convinced that the authors can show that these sugars could have been delivered to Earth. The two studied chondrites do not contain the same sugars (or deoxysugars) as the one formed in ice experiments. This is interpreted as the low stability of these compounds compared to their alcohol or acid derivative; I agree with that. Then, there is a glitch in the reasoning of the paper. If sugars and deoxysugars were synthesized in the ice grains that were accreted onto the chondrite parent bodies, secondary processes would have destroyed them, hence preventing the delivery of these sugars and deoxysugar at the surface of the primitive Earth. The conclusion drawn at lines 217–219 is then wrong. Or the authors have a mechanism to form sugars from their more stable

alcohol or acid derivatives on the primitive Earth. This would have to be supported by experiments, not performed here, or by references to other studies. I do not think that DNA or RNA could form with sugars not being in their aldose forms.

We modified the text in the manuscript regarding the topic of extraterrestrial delivery of organics, as mentioned in our answer to Reviewer #1's comment 1. The extraterrestrial delivery of organic material via meteorites is one plausible, global scenario for the origin of organics that may have been involved in the origin of life. Meteorites do not only bring sugar derivatives to the Earth and other planets, they also bring other organics, some of which are used in present-life (nucleobases, amino acids, etc.). Of course, new organics must have also been formed on the early Earth, so it is likely that organics important for life were both delivered from space via meteorites and formed on Earth. All these scenarios could plausibly have played key roles. Therefore, we are simply showing that our results are *consistent* with an extraterrestrial delivery scenario.

Regarding the survival of sugars during the formation of the Solar System and the fact that meteorites contain only mostly sugar alcohols and sugar acids, there are a few points we can develop here to show that the facts that sugars are less stable than their alcohol and acid counterparts and that meteorites mostly only deliver(ed) sugar alcohols and sugar acids does not necessarily mean that extraterrestrial delivery is not the origin of the sugars used in modern biology.

First, the facts that sugars are not as stable and that they were delivered as alcohols and acids may be a good thing, because this way they could survive until meteorites reach the Earth's surface. The oxidation of sugar alcohols and/or reduction of sugar acids into sugars may have happened at any or several later stages of life's early evolution. For example, once protected from the outside environment after organics were incorporated into the first proto-cells, life may have converted the ribitol and/or ribonic acid that were delivered via meteorites into ribose, because the sugar form appeared to be a better choice for a stable polymer.

Second, organics used by primitive life may be different from the materials used in modern biology. Several studies show that alternative polymers, e.g., glycol nucleic acids (GNA; Zhang et al. 2005, *J. Am. Chem. Soc.*, **127**, 4174), peptidic nucleic acid (PNA; Nielsen 2007, *Chem. Biodivers.*, **4**, 1996), or threose nucleic acid (TNA; Yu et al. 2012, *Nature Chemistry*, **4**, 183) may have played the role of primitive genetic materials and preceded RNA, i.e., without the use of ribose or any other sugar.

Third, meteoritic sugar acids possess D enantiomeric excesses (Cooper & Rios, 2016, *Proc. Natl. Acad. Sci.*, **113**, E3322) that they probably acquired in astrophysical environments. Although the origin of such excesses is still unknown, they may be related to the fact that modern RNA and DNA are exclusively made of D-ribose and D-deoxyribose, respectively.

Finally, meteorites are not the only means of delivery of organic material on Earth, as such organics can also be delivered via cometary grains and interplanetary dust particles (IDPs). According to some studies, the organic composition of cometary dust may likely be a better analogue of the material we produce in our experiments (see, e.g., Nuevo et al., 2011, *Adv. Space Res.*, **48**, 1126). Unfortunately, only a few cometary grains collected by the NASA Stardust mission could be analysed with the same techniques used for the analysis of laboratory samples, so it is difficult to draw general conclusions about these comparisons.

3. My main last concern with this manuscript is the connection made with the old scenario that building blocks of life on Earth were brought from space. This is not new and the literature about extraterrestrial organic matter is full of similar statements. In the end, this statement is really

overrated. The main criticism one may argue to this hypothesis is the concentration of these organic compounds in terrestrial prebiotic lakes or ocean. Even considering that 100% of organic compounds would survive the fall on Earth, the amount delivered would be too small to achieve a concentration high enough to induce chemical reactions similar to biochemical ones. And if we suppose repetitive falls of comets or meteorites at a specific location, to increase the quantity of delivered organic matter, then the energy resulting from these impacts would actually hamper the lifetime of organic molecules. The concern of the concentration of extraterrestrial organic compounds at the surface of the prebiotic Earth is never really discussed in the exobiology literature and this manuscript does not really care about it. For me, making the direct conclusion that, because these sugars could be produced by some processes in space, the building blocks of the first DNA molecules on Earth (or RNA and other biomolecules) are of extraterrestrial origin is the main weakness of the rationale of this study (see the beginning of the abstract).

Please refer to our previous answer. We have changed the text to make it clear that extraterrestrial delivery could have played a key role, while is unlikely to be solely responsible. However, it should be kept in mind the sheer volume of material (much of it carbonaceous) that enters Earth via extraterrestrial delivery. According to a recent study, even today the amount of new cosmic dusts entering Earth's atmosphere is estimated at ~60 tons/day (Gardner et al., 2014, *J. Geophys. Res.*, **119**, 7870). If we look back to 4.4 Gyr ago, this amount is estimated to be ~10⁹ kg of carbon/year from IDPs alone (i.e, not counting comets and meteorites; Chyba & Sagan, 1992, *Nature*, **355**, 125). In any event, our experiments demonstrate that extraterrestrial chemistry is expected to have created molecules of astrobiological importance.

A few minor comments:

4. At line 84 it is said that "All the residues produced in this work contained a wide variety of sugars, sugar alcohols, and sugar acids, with a similar distribution to previous laboratory studies" in addition to which they detected deoxysugar derivatives reported in table 1. I recommend to report the concentration of all the sugar related compounds, including compounds already reported in previous studies. That would be useful for comparison with future experiments.

As mentioned in some of the answers to Reviewers #1 and #2, we are currently working on a more detailed study of the formation of such sugar derivatives from the UV irradiation of ice mixtures of different compositions. Adding this information to the current manuscript will only make it much longer and more confusing, as we only focus on the formation of deoxysugar derivatives here. The only exception is ribose, for which we reported its abundances in the same residues in order to compare them to those of deoxyribose and use this information to discuss the formation mechanism of deoxyribose.

5. At line 115: it is said that yields are very variable and later it is said that the abundance of products in ¹³C-labelled experiments is lower. Why is that? How reliable are the yields reported here if there is so much variation?

As explained in great detail in the answer to Reviewer #1's comment 2, the chemistry taking place in these experiments does *not* follow the classical rules of thermodynamics. Consequently, two UV irradiation experiments carried out with the same ice composition, temperature, pressure,

irradiation time, etc., may not lead to fully identical results. Please also refer to our answer to Reviewer #2's comment 3a regarding the reason why this point was not discussed in the Meinert et al. (2016) paper.

6. As a summary of my review, this is a robust experimental study that should be submitted elsewhere, in a more specialized journal. The authors could then describe into more details the processes occurring in their ice residue, discuss (potentially new?) mechanisms for formation of complex molecules in space and soften a lot the connection between their data and the influence of extraterrestrial organic matter on the emergence of life.

We hope that the explanations we provided will convince the Reviewer that the results presented in our manuscript can reach the broad audience targeted by *Nature Communications*, and why the question of the formation mechanism(s) cannot be answered in this manuscript.

Reviewer #1 (Remarks to the Author):

The manuscript entitled 'Deoxyribose and deoxysugar derivatives in residues produced from photoprocessed astrophysical ice analogues and comparison to meteorites' deals with the formation of 2-deoxyribose and other deoxysugar derivatives in residues by ultraviolet irradiation of ice mixtures consisting of H₂O and CH₃OH.

The results reported in this manuscript are of broad interest and of great importance. Therefore I recommend publication in Nature Communications.

Here are my comments about the revised manuscript:

The authors addressed the discussion about organic molecules delivered by extraterrestrial objects and the possibility to form more complex organic matter on earth.

The authors are right, that it is not possible to distinguish epimers, but still it would be very nice to see a high-resolution MS/MS spectrum of the potentially formed deoxyribose for absolute confirmation of the structure. Our experience is that the fragmentation pathways of the diastereoisomers of this compound class shows some differences. The question was not aiming at the absolute or relative configuration, but on the structure itself.

The discussion about the formation of sugars via formose reaction is comprehensive and all points are considered in the revised text.

I recommend deleting the term 'co-catalyst' from page 9 in terms of reaction kinetics in such a complex reaction network. I am not absolutely sure, if glycolaldehyde, glyceraldehyde etc. are co-catalysts, which implies that there is an autocatalytic behavior. This has been proposed by Breslow and others, however a detailed kinetic analysis has been never performed. These higher order reaction get accelerated in the course of the reaction, because concentrations increase. The typical 's-shaped' curve for autocatalytic reactions are not observed.

Oliver Trapp

Department of Chemistry

Reviewer #2 (Remarks to the Author):

I have read the revised manuscript by Nuevo et al. in addition to their responses to the three reviewers. I still maintain the opinion that their research findings of deoxyribose and deoxysugar derivatives in multiple astrophysical ice analog residues is important enough to be published in Nature Communications. In my opinion, the authors adequately address questions in their "Response to Referees Letter"; however, many of these answers have not made it into the main text or supplementary information section – which I think is a mistake. I will leave it up to these very experienced authors whether they want to take my suggestions below and makes these additional edits (I hope they do though!)

1. I had asked earlier what is the reason for the range of abundances for deoxysugars in residues. The authors gave a clear, detailed, and plausible answer back in their responses to reviewers only. For example, one of the reasons for the differences in abundances was due to the stochastic nature of these experiments. However, the main text discusses stochastic effects in terms of considerations of mechanism and never addresses the effect it could have on abundances. The fact that there is a range of abundances that the author report, it should have some explanation either in the main text or supplementary information section. If Nature Communications targets a broad audience, then these details should be stated, which probably is known only to specialists.

2. I had asked earlier about trends in abundances. The authors supplied a thoughtful answer to reviewers, but only one sentence made its way into the supplementary info section regarding the GC-MS method that does not allow for the detection of smaller deoxysugar derivatives. There is no mention regarding the likely possibility of sublimation of lower molecular weight species in general biasing abundances. This seems like a worthwhile point to make in the supplementary text as well as how some lower molecular weight species can be detected in the first place if there are issues (the authors offer trapping within the refractory residue as one possibility).

3. I had asked earlier about the effect of NH₃ in these experiments based on a past paper by de Marcellus et al. The authors supplied a thoughtful answer to reviewers, but again, it did not make its way into the main or supplementary text. This information is important to include somewhere since

it implies that NH₃ or any NH₃-derived intermediate/product does not mediate these reactions. Otherwise we are left only with the conclusions supplied by de Marcellus et al. and it would seem like this is not the case anymore.

4. The authors provide an explanation for retention time shifts with some data, i.e. comparative samples not injected on the same day, and sometimes weeks or months apart. I would recommend including this note in the supplementary text.

5. From my understanding, Reviewer #3 (I'm Reviewer #2) had some concerns that sugar alcohols or acid derivatives (rather than sugars) would be delivered to early Earth based on the suggestions of the authors. The authors suggest scenarios of these compounds could have served as primitive alternatives before the modern compounds were adopted. Second, authors suggest that cometary dust may be a better analog to these experiments, therefore comets could be another source. It would be good to state these points somewhere. When differences in composition between ice experiments and meteorites arise, one, I instantly think of the effect of secondary processing in asteroids, and two, I think that delivery of cometary material may be more important than delivery of meteorites for these types of organics. The latter point may be somewhat speculative based on limited understanding of cometary ices, but I think that only reinforces the need for these astrophysical ice experiments even more.

6. There are a few examples of photoinduced formose reactions in the literature. In their point-by-point response the authors mentioned they could not find these papers, so I give one example with full citation information below. For example, see Olga A. Snytnikova, Alexandr N. Simonov, Oxana P. Pestunova, Valentin N. Parmon and Yuri P. Tsentalovich, Study of the photoinduced formose reaction by flash and stationary photolysis, *Mendeleev Commun.*, 2006, 16(1), 9–11. While the experiments in this paper take place in aqueous solutions (as opposed to extremely low temperature ices), it shows that condensation of formaldehyde into more complex aldehydes (glycolaldehyde and glyceraldehyde) and monosaccharides (glucose, lyxose, erythrose and erythrulose) takes place under UV irradiation in the absence of catalysts and initial primers (which has some commonalities with astrophysical ice analog experiments – namely UV light and no base catalyst). The observations of both sugars/derivatives (along with deoxysugars/derivatives) in the same ice residues suggests the possibility of a photoinduced formose reaction that may have had extra steps to get to the deoxysugars/derivatives. Although I agree with the authors that any discussion of mechanism is purely speculative at this stage without further evidence and is not needed for this particular paper.

Reviewer #3 (Remarks to the Author):

Nuevo et al. have addressed several comments in this revised version. I appreciate that they have softened the connection between organic matter in meteorites and the origin of life on Earth throughout the manuscript. Nevertheless, after reading the revised version and the rebuttal letter, I still have some comments/concerns that need to be addressed before acceptance for publication in Nature Communications.

1- The authors mention Cooper et al. 2001 in Nature to give an example of a paper dealing with a similar topic that was published in a broad audience journal. As a matter of facts, the aforementioned paper provided the structure of the detected molecules in a table/figure. I believe this could enhanced the present manuscript to show the structure of at least the molecules reported in table 1.

2-Although I understand the point made by the authors, I regret that the concentrations of all the sugar related compounds formed in the 5 experiments performed in this study are not reported. That would really constitute valuable data for the reader and allow the comparison with other ice experiment studies. If such data cannot fit within the main paper, this could constitute a table in the supplementary online material.

3-The authors now discuss potential processes to account for the formation of the detected sugars in ice experiments. Although it remains speculative due to high complexity of mechanisms occurring in the ice, I think this is a good point to discuss in the manuscript. However, there are still a few concerns.

3a-The starting ice in Meinert et al. 2016, the other study that has reported the formation of ribose in ice irradiation experiment, includes ammonia while in this study the starting ice only contains water and methanol. In many experiments, ammonia has a strong influence on the reaction pathways and their products. Can the author comment on this and discuss the influence of the presence/absence of NH₃ in the formation of ribose vs deoxyribose (and related sugars)?

3b- The authors state that UV irradiation of methanol/water ice results in a limited abundance of formaldehyde. This is partly incorrect. Abou Mrad et al. 2016 (Methanol ice VUV photo-processing: GC-MS analysis of volatile organic compounds, Monthly Notices of the Royal Astronomical Society 458(2):stw346) and 2017 (The Gaseous Phase as a Probe of the Astrophysical Solid Phase Chemistry, ApJ 846:124) have shown that formaldehyde is one of the major compounds formed by UV irradiation of ice containing methanol.

3c- A final step that may influence the synthesis processes is the final warming of the ice to recover the products. It has been shown by others, for instance by Theulé P., et al. 2013 (Thermal reactions in interstellar ice: a step towards molecular complexity in the interstellar medium, Advanced Space Research, 52 (8)), that active chemistry happens during the warming of the ice. Some products could form even in ice that did not receive any UV photons. How could this process influence the

formation of sugars and their derivatives in the present study? Could the authors provide the rate of heating in their experiments?

Reviewer #1 (Remarks to the Author):

The manuscript entitled 'Deoxyribose and deoxysugar derivatives in residues produced from photoprocessed astrophysical ice analogues and comparison to meteorites' deals with the formation of 2-deoxyribose and other deoxysugar derivatives in residues by ultraviolet irradiation of ice mixtures consisting of H₂O and CH₃OH.

The results reported in this manuscript are of broad interest and of great importance. Therefore I recommend publication in *Nature Communications*.

We thank again the Reviewer for recommending our manuscript for publication in *Nature Communications*.

Here are my comments about the revised manuscript:

1. The authors addressed the discussion about organic molecules delivered by extraterrestrial objects and the possibility to form more complex organic matter on earth.

We thank the Reviewer for acknowledging our answer and the changes we made.

2. The authors are right, that it is not possible to distinguish epimers, but still it would be very nice to see a high-resolution MS/MS spectrum of the potentially formed deoxyribose for absolute confirmation of the structure. Our experience is that the fragmentation pathways of the diastereoisomers of this compound class shows some differences. The question was not aiming at the absolute or relative configuration, but on the structure itself.

Unfortunately, our GC-MS instruments do not have the capability of measuring high-resolution MS/MS spectra, so it will not be possible to address the point raised by the Reviewer in the present study. The identification of the compounds is based on the comparison of both their retention times in the chromatograms and their mass spectra (fragmentation patterns) after 70-eV electron-impact ionization with commercial standards prepared following the exact same protocol. Given the facts that: (i) only two compounds (with two peaks each, corresponding to enantiomers) were shown to display mass spectra similar to 2-deoxyribose, (ii) these compounds have very different retention times, and (iii) the retention time and the mass spectrum of the peak assigned to 2-deoxyribose in the samples matches very well those of the deoxyribose standard, we do not see any reason to further confirm the structure of this compound. In addition, experiments in which the starting methanol was labelled with ¹³C, show that all products are fully isotopically marked with ¹³C. Therefore, we are confident about the assignments of the compounds identified in our residues and listed in Table 1 of the manuscript.

In the case the Reviewer was suggesting to use a high-resolution MS/MS technique to better characterize the structure of the compound that we tentatively assigned to 2-deoxyxylose, and confirm its identification, we still think such technique would not be sufficient because there is no standard of 2-deoxyxylose to which we could compare the high-resolution MS/MS spectra from the residues. If the reviewer meant that high-resolution MS/MS could help determining whether the peaks tentatively identified as 2-deoxyxylose are instead due to another deoxyribose compound (i.e., 3- or 4-deoxyribose), the problem would be the same without running standards

of these compounds, which are not commercially available. Moreover, 3- and 4-deoxyribose are expected to show different fragmentation patterns compared with 2-deoxyribose as well as 2-deoxyxylose. Therefore, the fact that the mass spectra of the peaks around 57 min in our residues (Fig. S2) and that of 2-deoxyribose (Fig. 1) show strong similarities supports the assignment of these peaks to 2-deoxyxylose, even if it does not definitely prove it, hence the reported tentative identification.

Finally, if the Reviewer was referring to the conformation (linear vs. cyclic) of 2-deoxyribose (and maybe 2-deoxyxylose?), we did not look into this particular aspect, and focused mainly on the identification of this compound via direct comparison with the commercial standard prepared under the same conditions, regardless of its conformation in the solutions injected into the GC-MS device. However, decades of studies of 5C and 6C sugar and deoxysugar derivatives have shown that such compounds are dominantly in their cyclic form when put in solution, with less than 1% of opened-chains when put in water.

3. The discussion about the formation of sugars via formose reaction is comprehensive and all points are considered in the revised text.

We thank the Reviewer for acknowledging our explanations and the changes made to the manuscript regarding this point.

4. I recommend deleting the term ‘co-catalyst’ from page 9 in terms of reaction kinetics in such a complex reaction network. I am not absolutely sure, if glycolaldehyde, glyceraldehyde etc. are co-catalysts, which implies that there is an autocatalytic behavior. This has been proposed by Breslow and others, however a detailed kinetic analysis has been never performed. These higher order reaction get accelerated in the course of the reaction, because concentrations increase. The typical ‘s-shaped’ curve for autocatalytic reactions are not observed.

We understand that the term “co-catalyst” is confusing here. Since the terms “initiator” and “initial primer” are also often used when referring to the role of these compounds in formose-type reactions, **we decided to replace the term “co-catalyst” by “initiator”**.

Oliver Trapp
Department of Chemistry
Ludwig-Maximilians-University Munich

Reviewer #2 (Remarks to the Author):

I have read the revised manuscript by Nuevo et al. in addition to their responses to the three reviewers. I still maintain the opinion that their research findings of deoxyribose and deoxysugar derivatives in multiple astrophysical ice analog residues is important enough to be published in *Nature Communications*. In my opinion, the authors adequately address questions in their “Response to Referees Letter”; however, many of these answers have not made it into the main text or supplementary information section – which I think is a mistake. I will leave it up to these very experienced authors whether they want to take my suggestions below and makes these additional edits (I hope they do though!).

1. I had asked earlier what is the reason for the range of abundances for deoxysugars in residues. The authors gave a clear, detailed, and plausible answer back in their responses to reviewers only. For example, one of the reasons for the differences in abundances was due to the stochastic nature of these experiments. However, the main text discusses stochastic effects in terms of considerations of mechanism and never addresses the effect it could have on abundances. The fact that there is a range of abundances that the author report, it should have some explanation either in the main text or supplementary information section. If *Nature Communications* targets a broad audience, then these details should be stated, which probably is known only to specialists.

The Reviewer made a good point, so we added a sentence in the “Constraints on the formation mechanisms” section of the manuscript to say that abundances can vary significantly from one sample to another due to the stochastic nature of the processes involved in these experiments, even though they were produced under similar conditions and from similar starting ice mixtures.

2. I had asked earlier about trends in abundances. The authors supplied a thoughtful answer to reviewers, but only one sentence made its way into the supplementary info section regarding the GC-MS method that does not allow for the detection of smaller deoxysugar derivatives. There is no mention regarding the likely possibility of sublimation of lower molecular weight species in general biasing abundances. This seems like a worthwhile point to make in the supplementary text as well as how some lower molecular weight species can be detected in the first place if there are issues (the authors offer trapping within the refractory residue as one possibility).

We added a sentence in the “Constraints on the formation mechanisms” section of the manuscript to explain why some small compounds may be either lost by sublimation during the warm-up stage, or trapped in the residues recovered at room temperature with the rest of the photoproducts.

3. I had asked earlier about the effect of NH_3 in these experiments based on a past paper by de Marcellus et al. The authors supplied a thoughtful answer to reviewers, but again, it did not make its way into the main or supplementary text. This information is important to include somewhere since it implies that NH_3 or any NH_3 -derived intermediate/product does not mediate these reactions. Otherwise we are left only with the conclusions supplied by de Marcellus et al. and it would seem like this is not the case anymore.

Since none of our starting mixtures contained any NH_3 , it would be irrelevant to discuss the effects of a compound that is not involved in any of the reactions and product formation described in our manuscript. For this reason, the effect of NH_3 in the formation of sugar derivatives will not be covered in this manuscript. However, our study clearly shows that NH_3 is not needed to form deoxysugar (and sugar) derivatives. The effect of NH_3 on the formation of polyols will be covered in more detail in an upcoming manuscript that will summarize results obtained for more than 20 samples, some of which were produced from the UV irradiation of ices containing NH_3 .

4. The authors provide an explanation for retention time shifts with some data, i.e. comparative samples not injected on the same day, and sometimes weeks or months apart. I would recommend including this note in the supplementary text.

This is also a good point, and as suggested by the Reviewer, **we added two sentences in the Supplementary Information to add this piece of information.**

5. From my understanding, Reviewer #3 (I'm Reviewer #2) had some concerns that sugar alcohols or acid derivatives (rather than sugars) would be delivered to early Earth based on the suggestions of the authors. The authors suggest scenarios of these compounds could have served as primitive alternatives before the modern compounds were adopted. Second, authors suggest that cometary dust may be a better analog to these experiments, therefore comets could be another source. It would be good to state these points somewhere. When differences in composition between ice experiments and meteorites arise, one, I instantly think of the effect of secondary processing in asteroids, and two, I think that delivery of cometary material may be more important than delivery of meteorites for these types of organics. The latter point may be somewhat speculative based on limited understanding of cometary ices, but I think that only reinforces the need for these astrophysical ice experiments even more.

We agree with the Reviewer that the delivery of cometary materials could have significantly contributed to the inventory of organics that played a role in very early biology. However, we would like to point out that the fact that discrepancies between the distribution of the sugar and deoxysugar derivatives found in residues and that in meteorites does not necessarily mean that such experimental simulations do not reproduce processes taking place in meteorites, or that secondary processes (aqueous alteration) must account for the discrepancies. One of the most obvious differences between our experiments and processes taking place in astrophysical environments is the composition of the starting ice mixtures. The starting mixtures ($\text{H}_2\text{O}:\text{CH}_3\text{OH}$) studied in our present experiments constitute a simple model of ices which are known to be present in abundance in cold astrophysical environments, and which are believed to play an important role in the formation of sugar and deoxysugar derivatives in these environments.

These experiments show that such organics can indeed be formed from the UV irradiation of $\text{H}_2\text{O}:\text{CH}_3\text{OH}$ ice mixtures at low temperature. However, the presence of other species in the starting ices, including NH_3 (as briefly discussed above in Comment 3 and in our last Response to Reviewers), as well as other carbon sources known to be present in astrophysical ices such as CO , CO_2 , HCN , H_2CO , etc., may affect the final distribution of organics formed in such experiments. The effect of NH_3 and some of these additional carbon sources will be discussed in more detail in the upcoming manuscript we mentioned above (see Comment 3), but we would note that the distribution of sugar derivatives in the unique residue analysed by Meinert et al. (2016), which was

produced from the UV irradiation of an H₂O:CH₃OH:NH₃ ice mixture, does not match the meteoritic distribution either (see Cooper et al. 2001), and rather matches the distribution we observed in our residues produced from ice mixtures that did not contain any NH₃.

Therefore, comparing our experiments of UV irradiation of simple H₂O:CH₃OH ice mixtures to meteoritic samples is scientifically relevant, but it unfortunately does not tell the whole story. More experiments involving different ice mixtures relevant to astrophysical environments as well as studies of aqueous alteration of the resulting residues will provide a better understanding of these processes and allow for a better comparison with meteoritic and cometary materials.

In addition, little is known of the organic composition of comets, with the exception of the bulk composition of cometary samples returned from comet 81P/Wild 2 by the Stardust mission (e.g., Cody et al., 2008, *Meteorit. Planet. Sci.*, **43**, 353; Elsila et al., 2009, *Meteorit. Planet. Sci.*, **44**, 1323) and the organic compounds identified in the coma and on the surface of comet 67/Churyumov-Gerasimenko by the Rosetta mission (e.g., Wright et al., 2015, *Science*, **349**, aab0673; Goesmann et al., 2015, *Science*, **349**, aab0689; Altwegg et al., 2016, *Sci. Adv.*, **2**, 1600285). To the best of our knowledge, the most detailed comparison between the organic composition of cometary materials and laboratory residue to date is the study performed by our group of residues produced from the UV irradiation of ice mixtures of compositions H₂O:CH₃OH:CO:NH₃ with and without the addition of C₃H₈ (propane) or naphthalene (C₁₀H₈), i.e., mixtures more complex than those of the residues studied in the present manuscript, and their comparison to cometary materials using X-ray absorption near-edge structure (XANES) spectroscopy (Nuevo et al., 2011, *Adv. Space Res.*, **48**, 1126).

Because of the lack of data for cometary materials, in particular the lack of data regarding the search for complex organics such as sugar derivatives, we did not modify the text in the manuscript, as such a statement would be too speculative.

6. There are a few examples of photoinduced formose reactions in the literature. In their point-by-point response the authors mentioned they could not find these papers, so I give one example with full citation information below. For example, see Olga A. Snytnikova, Alexandr N. Simonov, Oxana P. Pestunova, Valentin N. Parmon and Yuri P. Tsentelovich, Study of the photoinduced formose reaction by flash and stationary photolysis, *Mendeleev Commun.*, 2006, **16(1)**, 9–11. While the experiments in this paper take place in aqueous solutions (as opposed to extremely low temperature ices), it shows that condensation of formaldehyde into more complex aldehydes (glycolaldehyde and glyceraldehyde) and monosaccharides (glucose, lyxose, erythrose and erythrulose) takes place under UV irradiation in the absence of catalysts and initial primers (which has some commonalities with astrophysical ice analog experiments – namely UV light and no base catalyst). The observations of both sugars/derivatives (along with deoxysugars/derivatives) in the same ice residues suggests the possibility of a photoinduced formose reaction that may have had extra steps to get to the deoxysugars/derivatives. Although I agree with the authors that any discussion of mechanism is purely speculative at this stage without further evidence and is not needed for this particular paper.

We thank the Reviewer for providing the full reference information of that paper which is very interesting and should be mentioned in our manuscript. Therefore, **we added a sentence saying that UV irradiation may help initiating a formose-type reaction without the presence of any basic catalyst or initiator, as described in Snytnikova et al. (2006).**

Reviewer #3 (Remarks to the Author):

Nuevo et al. have addressed several comments in this revised version. I appreciate that they have softened the connection between organic matter in meteorites and the origin of life on Earth throughout the manuscript. Nevertheless, after reading the revised version and the rebuttal letter, I still have some comments/concerns that need to be addressed before acceptance for publication in *Nature Communications*.

We hope these extra revisions will address the last concerns of the reviewers. Our detailed responses to his/her comments and questions are given below.

1. The authors mention Cooper et al. 2001 in *Nature* to give an example of a paper dealing with a similar topic that was published in a broad audience journal. As a matter of fact, the aforementioned paper provided the structure of the detected molecules in a table/figure. I believe this could enhance the present manuscript to show the structure of at least the molecules reported in table 1.

We agree that showing the molecular structures of these compounds is useful. We did think about adding a separate figure showing the molecular structures of all the compounds identified in our samples, or even all the compounds searched for in this study. However, since the manuscript already contains quite a number of figures and tables, we decided instead to add the molecular structures of the identified compounds embedded in the figures showing the mass spectra of the standards, as can be seen in all versions (original and revised) of the manuscript. For instance, the bottom panel of Fig. 1b shows the mass spectrum of a 2-deoxyribose standard, together with its molecular structure (on the mass spectrum plot itself), that is compared to the mass spectra of the peaks corresponding to 2-deoxyribose in the residues. We did the same thing in all figures showing the identification or tentative identification of deoxysugar derivatives.

2. Although I understand the point made by the authors, I regret that the concentrations of all the sugar related compounds formed in the 5 experiments performed in this study are not reported. That would really constitute valuable data for the reader and allow the comparison with other ice experiment studies. If such data cannot fit within the main paper, this could constitute a table in the supplementary online material.

Adding this information to the manuscript will make it significantly longer and would divert the readers from the main focus of this study, i.e., the identification of deoxysugar derivatives. Moreover, it is objectively impossible to add a table showing the identification of sugar derivatives and their abundances without adding an entirely new discussion section, even if such a table is added to the Supplementary Information. As we mentioned briefly in our previous Response to Reviewers, as well as in the response to Reviewer #2's comment 3 above, we are currently working on a manuscript that will include results obtained for more than 20 samples, and that will include a list of all the sugar derivatives identified in these residues together with their abundances.

3. The authors now discuss potential processes to account for the formation of the detected sugars in ice experiments. Although it remains speculative due to high complexity of mechanisms

occurring in the ice, I think this is a good point to discuss in the manuscript. However, there are still a few concerns.

3a. The starting ice in Meinert et al. 2016, the other study that has reported the formation of ribose in ice irradiation experiment, includes ammonia while in this study the starting ice only contains water and methanol. In many experiments, ammonia has a strong influence on the reaction pathways and their products. Can the author comment on this and discuss the influence of the presence/absence of NH_3 in the formation of ribose vs deoxyribose (and related sugars)?

As mentioned in our response to Reviewer #2's comment 3, discussing the effect of NH_3 would be irrelevant for the present study, as no NH_3 was used in any of the starting mixtures. This topic will be covered in more detail in an upcoming manuscript.

3b. The authors state that UV irradiation of methanol/water ice results in a limited abundance of formaldehyde. This is partly incorrect. Abou Mrad et al. 2016 (Methanol ice VUV photoprocessing: GC-MS analysis of volatile organic compounds, *Monthly Notices of the Royal Astronomical Society* 458(2):346) and 2017 (The Gaseous Phase as a Probe of the Astrophysical Solid Phase Chemistry, *ApJ* 846:124) have shown that formaldehyde is one of the major compounds formed by UV irradiation of ice containing methanol.

As mentioned in the two papers mentioned by the Reviewer, as well as in other studies, formaldehyde is indeed a well-known and abundant photolysis product of methanol. However, one should keep in mind that the abundances of the products detected by Abou Mrad et al. were measured at room temperature after warm-up. Therefore, it is difficult to tell how much formaldehyde is actually formed and present in the ices at low temperature, available for reactions, before warm-up. In low-temperature ices, methanol's photoproducts are numerous and, besides formaldehyde, also include CO , CO_2 , CH_2OH (radical), CH_3O (radical), and CH_3+OH (radicals), as well as ionic variants of these species. Among these, radicals and ions may have important roles in the formation of complex organics, because reactions with these species have very low to virtually no activation barriers. However, since they cannot be detected at room temperature during warm-up the same way as formaldehyde, it is difficult to determine their abundances in the ices compared with formaldehyde, and how important their role is in the formation of complex organics such as sugar derivatives. Nonetheless, **we modified one sentence in the manuscript to reinforce the fact that formaldehyde is one of the major photoproducts of methanol to strengthen this point from the Reviewer's comment.** We also added one of the references mentioned by the reviewer (Abou Mrad et al. 2016), but we could not add the other one (Abou Mrad et al. 2017), as we already reached the maximum of references we could cite in the manuscript (50).

3c. A final step that may influence the synthesis processes is the final warming of the ice to recover the products. It has been shown by others, for instance by Theulé P., et al. 2013 (Thermal reactions in interstellar ice: a step towards molecular complexity in the interstellar medium, *Advanced Space Research*, 52(8)), that active chemistry happens during the warming of the ice. Some products could form even in ice that did not receive any UV photons. How could this process influence the formation of sugars and their derivatives in the present study? Could the authors provide the rate of heating in their experiments?

We fully agree on the fact that chemistry does happen during the warm-up step. In addition to the paper mentioned by the Reviewer, this was also shown in a previous study describing the formation of carbamic acid (NH_2COOH) after UV irradiation of $\text{H}_2\text{O}^{12/13}\text{CO}_2\text{:NH}_3$ ice mixtures and subsequent warm-up (Nuevo et al., 2007, *Astron. Astrophys.* **464**, 253). On the other hand, other studies have shown that complex organic molecules, some of which are structurally similar to sugar derivatives, also form at temperatures as low as 5 K (Maity et al., 2015, *Phys. Chem. Chem. Phys.* **17**, 3081). Therefore, due to the complexity and the stochastic nature of the processes taking place in these experiments, it is likely that sugar derivatives form both at very low temperature in the ices and during warm-up. **In the manuscript, we added a short sentence to mention that some products may also form and during warm-up from reactions between species formed at lower temperature in the ices.** Regarding the experiments described in the present study, samples were warmed up from 12 K to room temperature at about 0.75 K min^{-1} . **This warm-up rate was also added to the manuscript.**

Reviewer #1 (Remarks to the Author):

The results reported in this manuscript are of broad interest and of great importance. Therefore I recommend publication in Nature Communications.

The authors addressed all my points.

I understand that they are limited by instrumentation to provide HR-MS and HR-MS/MS spectra. The identity has been proven by reference samples.

I recommend accepting the manuscript in the present form.

Reviewer #2 (Remarks to the Author):

I have no further comments regarding this manuscript. The authors made many changes to address reviewers' comments. I would recommend publication. Nice work.

One last thing - the last sentence in the abstract reads strangely to me now ("Finally, the deoxysugar derivatives found in our residues are compared with previous and new from studies of carbonaceous meteorites.") You might want to take a second look and make sure that's what you meant.

Reviewer #3 (Remarks to the Author):

I've reviewed this 2nd revised version of the manuscript by Nuevo et al. and I think it is now suitable for publication. However, I still believe that the paper would be improved by adding a separate figure with the structure of the main molecules discussed here, as well as a table providing the concentration of all the sugar molecules detected in the 5 experiments described. Both constitute valuable input to such a paper. It is unfortunate that the authors are reluctant for this add-on.

One last suggestion, I think the authors should switch ref 6 (Sephton 2002) by Martins, Z., Sephton, M., 2009. Extraterrestrial Amino Acids, in: Hughes, A.B. (Ed.), Amino Acids, Peptides and Proteins in Organic Chemistry. Vol.1 – Origins and Synthesis of Amino Acids. Wiley VCH, Weinheim, pp. 3-42.

It is more appropriate (as this is a review focused on amino acids) and that won't hurt M. Sephton, author of both.

I'm looking forward to see this manuscript published in Nature Com. I'm now convinced it will constitute a notable contribution to the field.

Reviewer #1 (Remarks to the Author):

The results reported in this manuscript are of broad interest and of great importance. Therefore I recommend publication in *Nature Communications*.

The authors addressed all my points.

I understand that they are limited by instrumentation to provide HR-MS and HR-MS/MS spectra. The identity has been proven by reference samples.

I recommend accepting the manuscript in the present form.

Thank you.

Reviewer #2 (Remarks to the Author):

I have no further comments regarding this manuscript. The authors made many changes to address reviewers' comments. I would recommend publication. Nice work.

One last thing - the last sentence in the abstract reads strangely to me now ("Finally, the deoxysugar derivatives found in our residues are compared with previous and new from studies of carbonaceous meteorites."). You might want to take a second look and make sure that's what you meant.

We modified this last sentence into "Finally, we report that some of the deoxysugar derivatives found in our residues are also newly identified in carbonaceous meteorites."

Reviewer #3 (Remarks to the Author):

I've reviewed this 2nd revised version of the manuscript by Nuevo et al. and I think it is now suitable for publication. However, I still believe that the paper would be improved by adding a separate figure with the structure of the main molecules discussed here, as well as a table providing the concentration of all the sugar molecules detected in the 5 experiments described. Both constitute valuable input to such a paper. It is unfortunate that the authors are reluctant for this add-on.

In this final revised version of the manuscript, we have added a figure showing the molecular structures of all the compounds mentioned in Table 1, i.e., compounds identified and tentatively identified in our residues as well as those found in meteorites. Regarding the addition of a table with the abundances of all compounds identified in the 5 residues discussed in our study, although we saw several sugar derivatives in these residues, we have not calculated their abundances, and we focused solely on the deoxysugar derivatives in the current paper. An analysis concerning the sugars will be done for the upcoming paper we mentioned in our previous responses, but it will take more time (and space) than reasonable to do so in the publication of the present paper.

One last suggestion, I think the authors should switch ref 6 (Sephton 2002) by Martins, Z., Sephton, M., 2009. Extraterrestrial Amino Acids, in: Hughes, A.B. (Ed.), *Amino Acids, Peptides and Proteins in Organic Chemistry*. Vol.1 – Origins and Synthesis of Amino Acids. Wiley VCH, Weinheim, pp. 3-42. It is more appropriate (as this is a review focused on amino acids) and that won't hurt M. Sephton, author of both.

We have replaced the "Sephton 2002" reference by the "Martins & Sephton 2009" reference as suggested by the reviewer.

I'm looking forward to see this manuscript published in *Nature Com*. I'm now convinced it will constitute a notable contribution to the field.

Thank you.